# ExID: Offline RL with Intuitive Expert Insights in Limited-Data Settings

## Abstract

With the ability to learn from static datasets, Offline Reinforcement Learning (RL) emerges as a compelling avenue for real-world applications. However, state-of-the-art offline RL algorithms perform sub-optimally when confronted with limited data confined to specific regions within the state space. The performance degradation is attributed to the inability of offline RL algorithms to learn appropriate actions for rare or unseen observations. This paper proposes a novel domain knowledge-based regularization technique and adaptively refines the initial domain knowledge to considerably boost performance in limited data with partially omitted states. The key insight is that the regularization term mitigates erroneous actions for sparse samples and unobserved states covered by domain knowledge. Empirical evaluations on standard offline RL datasets demonstrate a substantial average performance increase compared to ensemble of domain knowledge and existing offline RL algorithms operating on limited data.

## 1 Introduction

Offline RL (Ernst et al., 2005; Pru, 2023), also referred to as batch RL, is a learning approach that focuses on extracting knowledge solely from static datasets. This class of algorithms has a wider range of applications being particularly appealing to real-world data sets from business (Zhang & Yu, 2021), healthcare (Liu et al., 2020), and robotics (Sinha et al., 2022). However, offline RL poses unique challenges, including over-fitting and the need for generalization to data not present in the dataset. To surpass the behavior policy, offline RL algorithms need to query Q values of actions not in the dataset, causing extrapolation errors (Kumar et al., 2019). Most offline RL algorithms address this problem by enforcing constraints that ensure that the learned policy does not deviate too far away from the data set's state action distribution (Fujimoto et al., 2019b; Fujimoto & Gu, 2021) or is conservative towards Out-of-Distribution (OOD) actions (Kumar et al., 2019; Kostrikov et al., 2021). However, such approaches are designed on coherent batches (Fujimoto et al., 2019b), which do not account for OOD states.

In many domains, such as business and healthcare, available data is scarce and often confined to expert behaviors within a limited state space. *For example, a sales recommendation system, where historic data may not contain details about many active users and operator gives coupon of higher value to attract sales.* Learning on such limited data sets can curtail the generalization capabilities of state-of-the-art (SOTA) offline RL algorithms, resulting in sub-optimal performance (Levine et al., 2020a). We illustrate this limitation via Fig 1. In Fig 1a) the state action space of a simple Mountain Car environment (Moore, 1990) is plotted for an expert dataset (Schweighofer et al., 2022) and a partial dataset with first 10% samples from the entire dataset. Fig 1b) shows the average reward obtained over these data sets and the average difference between the Q value of action taken by the under-performing Conservative Q Learning (CQL) (Kumar et al., 2019) agent and the action in the full expert dataset for unseen states. It can be observed that the performance of the offline RL agent considerably drops. This is attributed to the critic overestimating the Q value of non-optimal actions for states that do not occur in the dataset while training.

In numerous real-world applications, expert insights regarding the general behavior of a policy are often accessible (Silva & Gombolay, 2021). *For example, sales operators often distribute lower discount coupons to active users to maximize profit.* While these insights may not be optimal, they serve as valuable guidelines for understanding the overall behavior of the policy. A rich literature in

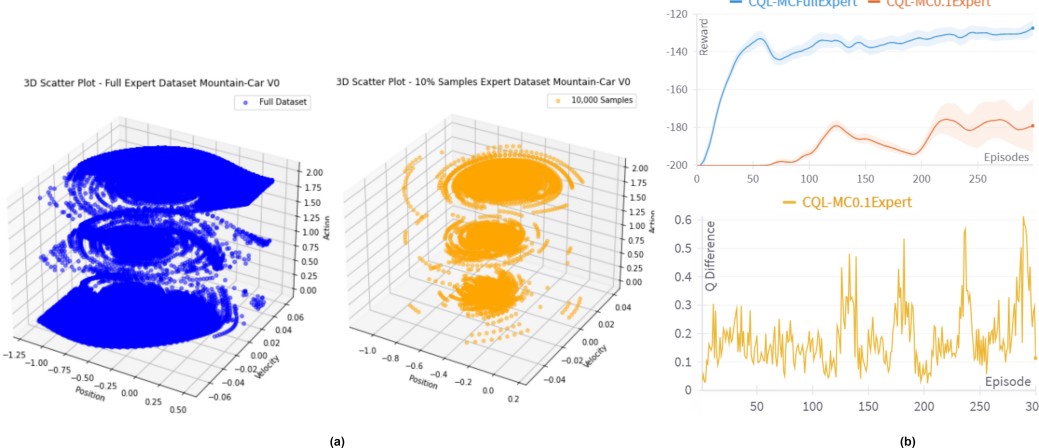

Figure 1: a) Full expert, Mountain Car dataset, and reduced dataset with first 10% samples showing distribution of state (position, velocity) and action b) CQL agent converging to a sub-optimal policy for reduced dataset exhibiting high Q values for actions different from actions in the expert dataset for unseen states.

knowledge distillation (Hu et al., 2016) has shown that teacher networks trained on domain knowledge can transfer knowledge to another network unaware of it. This work aims to leverage a teacher network mimicking simple decision tree-based domain knowledge to help offline RL generalize in limited data settings.

The paper makes the following novel contributions:

- We introduce an algorithm dubbed **ExID**, leveraging intuitive human obtainable expert insights. The domain expertise is incorporated into a teacher policy, which improves offline RL in limited-data settings through regularization.
- The teacher based on expected performance improvement of the offline policy during training, improving the teacher network beyond initial heuristics.
- We demonstrate the effectiveness of our methodology on *real world sales promotion dataset, simglucose dataset*, several OpenAI gym and Minigrid environments with standard offline RL data sets and show that ExID significantly exceeds the performance when faced with limited data.

## 2 RELATED WORK

This work improves offline RL learning on batches sampled from static datasets using domain expertise. One of the major concerns in offline RL is the erroneous extrapolation of OOD actions (Fujimoto et al., 2019b). Three techniques have been studied in the literature to prevent such errors. 1) Constraining the policy to be close to the behavior policy 2) Penalizing overly optimistic Q values (Levine et al., 2020b) 3) Learning model dynamics from data (Kidambi et al., 2020; Yu et al., 2020), where performance highly depends on the accuracy of the learned dynamics. We discuss a few relevant algorithms following these principles. In Batch-Constrained deep Q-learning (BCQ) (Fujimoto et al., 2019b) candidate actions sampled from an adversarial generative model are considered, aiming to balance proximity to the batch while enhancing action diversity. Algorithms like Random Ensemble Mixture Model (REM) (Agarwal et al., 2020), Ensemble-Diversified Actor-Critic (EDAC) (An et al., 2021) and Uncertainty Weighted Actor-Critic (UWAC) (Wu et al., 2021) penalize the Q value according to uncertainty by either using Q ensemble networks or directly weighting the loss with uncertainty. CQL (Kumar et al., 2019) enforces regularization on Q-functions by incorporating a term that reduces Q-values for OOD actions while increasing Q-values for actions within the expected distribution. However, these algorithms do not handle OOD actions for states not in the static dataset and can have errors induced by changes in transition probability.

Integration of domain knowledge in offline RL, though an important avenue, has not yet been extensively explored. Domain knowledge incorporation has improved online RL with tight regret bounds (Silva & Gombolay, 2021; Bartlett & Tewari, 2009). In offline RL, bootstrapping via blending heuristics computed using Monte-Carlo returns with rewards has shown to outperform SOTA algorithms by 9% (Geng et al., 2023). Recent works improve offline RL by incorporating a safety expert (Verma et al., 2024) and preference query (Yang et al., 2023), contrary to our work which improves imperfect domain knowledge. The closest to our work is Domain Knowledge guided Q learning (DKQ) (Zhang & Yu, 2021) where domain knowledge is represented in terms of action importance and the Q value is weighted according to importance. However, obtaining action importance in practical scenarios is nontrivial.

## 3 PRELIMINARIES

A DRL setting is represented by a Markov Decision Process (MDP) formalized as $(S, A, T, r, \rho_0, \gamma)$. Here, $S$ denotes the state space, $A$ signifies the action space, $T(s'|s, a)$ represents the transition probability distribution, $r : S \times A \rightarrow \mathbb{R}$ is the reward function, $\rho_0$ represents the initial state distribution, and $\gamma \in (0, 1]$ is the discount factor. The primary objective of any DRL algorithm is to identify an optimal policy $\pi(a|s)$ that maximizes $\mathbb{E}_{s_t, a_t}[\sum_{t=0}^{\infty} \gamma^t r(s_t, a_t)]$ where, $s_0 \sim d_0(.), a_t \sim \pi(.|s_t)$, and $s' \sim T(.|s_t, a_t)$. Deep Q networks (DQNs) (Mnih et al., 2015) learn this objective by minimizing the Bellman residual $(Q_\theta(s, a) - B^{\pi_\theta} Q_\theta(s, a))^2$ where $B^{\pi_\theta} Q_\theta(s, a) = \mathbb{E}_{s' \sim T}[r(s, a) + \gamma \mathbb{E}_{a' \sim \pi_\theta(.|s')}[Q_{\theta'}(s', a')]]$ where $\theta'$ is target network. The policy $\pi_\theta$ chooses actions that maximize the Q value $\max_{a' \in A} Q_\theta(s', a')$. However, in offline RL where transitions are sampled from a pre-collected dataset $\mathcal{B}$, the chosen action $a'$ may exhibit a bias towards OOD actions with inaccurately high Q-values. To handle the erroneous propagation from OOD actions, CQL (Kumar et al., 2020) learns conservative Q values by penalizing OOD actions. The CQL loss for critic network is given by

$$\mathcal{L}_{cql}(\theta) = \min_Q \alpha \, \mathbb{E}_{s \sim \mathcal{B}}[log \sum_a exp(Q_\theta(s, a)) -$$

$$\mathbb{E}_{a \sim \mathcal{B}|s}[Q_\theta(s, a)]] + \frac{1}{2}\mathbb{E}_{s,a,s' \sim \mathcal{B}}[Q_\theta - Q_{\theta'}]^2 \quad (1)$$

Eq. 1 encourages the policy to be close to the actions seen in the dataset. However, CQL works on the assumption of coherent batches, i.e., if $(s, a, s') \in \mathcal{B}$, then $s' \in \mathcal{B}$. There is no provision for handling OOD actions for $s \notin \mathcal{B}$, which can lead to policy failure when data is limited. In the next sections, we present ExID, a domain knowledge-based approach to improve performance in data-scarce scenarios.

## 4 PROBLEM SETTING AND METHODOLOGY

In our problem setting, the RL agent learns the policy on a limited dataset with rare and unseen demonstrations. We define the characteristics of this dataset as follows:

**Definition 4.1.** Let $\mathcal{B}$ be the original offline reinforcement learning buffer, represented as a multiset of transitions $(s, a, s')$. Each transition $(s, a, s')$ appears a certain number of times in $\mathcal{B}$, which we denote as $N_\mathcal{B}(s, a, s')$.

The reduced buffer $\mathcal{B}_r$ is a sub-multiset of $\mathcal{B}$, such that the number of occurrences of any transition $(s, a, s')$ in $\mathcal{B}_r$, denoted $N_{\mathcal{B}_r}(s, a, s')$, satisfies:

$$N_{\mathcal{B}_r}(s, a, s') \leq N_\mathcal{B}(s, a, s').$$

We observe, *performing $Q - Learning$ by sampling from a limited buffer $\mathcal{B}_r$ may not converge to an optimal policy for the MDP $M_\mathcal{B}$ representing the full buffer.* This can be shown as a special case of (Theorem 1,(Fujimoto et al., 2019b)) as $p_\mathcal{B}(s'|s, a) \neq p_{\mathcal{B}_r}(s'|s, a)$ and no Q updates for $(s, a) \notin \mathcal{B}_r$ leading to sub-optimal policy. Please refer to the App. B for analysis and example.

We assume that a set of common-sense rules in the form of domain knowledge, denoted as $\mathcal{D}$, is available. This domain knowledge defines a hierarchical mapping from states to actions ($S \rightarrow$

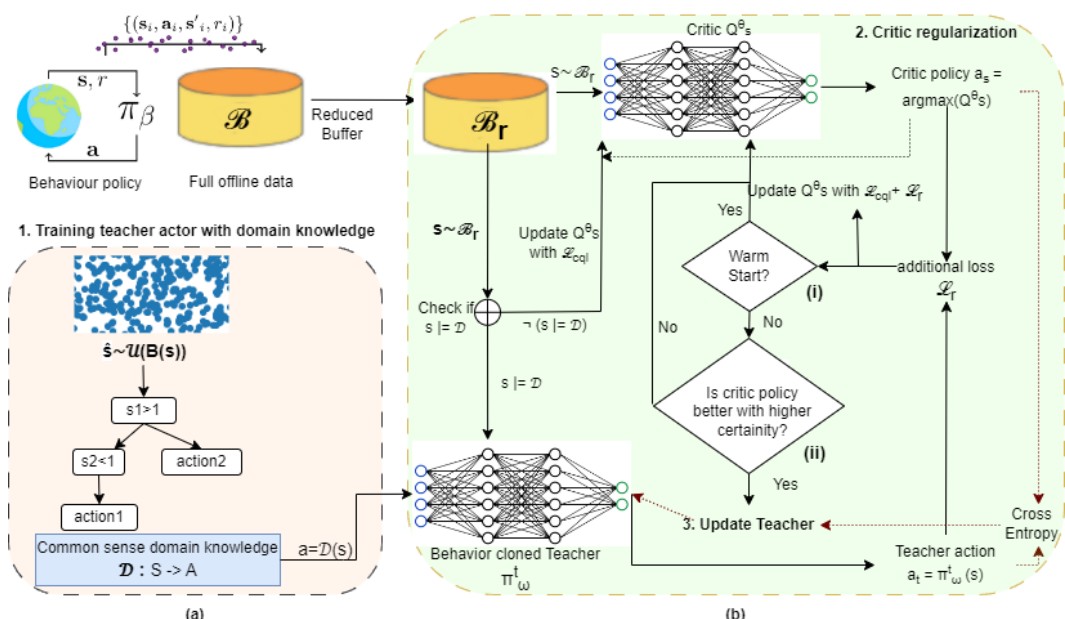

Figure 2: Overview of the proposed methodology (a) Training a teacher policy network with domain knowledge and synthetic data (b) Updating the offline RL critic network with teacher network

$A$), structured as decision nodes. Each decision node $T_{\eta_i}$ has constraint $\phi_{\eta_i}$ that determines its branching, a Boolean indicator $\mu_{\eta_i}$ selects the branch ($\swarrow$ or $\searrow$) to follow based on whether the constraint $\phi_{\eta_i}$ is satisfied.

$$Action = \begin{cases} a_{\eta_i} & \text{if } leaf \\ \mu_{\eta_i}T_{\eta_i\swarrow}(s) + (1-\mu_{\eta_i})T_{\eta_i\searrow}(s) & \text{o/w} \end{cases}$$
$$\mu_{\eta_i}(s) = \begin{cases} 1 & \text{if } s \models \phi_{\eta_i} \\ 0 & \text{o/w} \end{cases} \tag{2}$$

We assume that $\mathcal{D}$ gives heuristically reasonable actions for $s \models D$ and $S_{\mathcal{D}} \cap S_{\mathcal{B}_r} \neq \emptyset$ where $S_{\mathcal{D}}, S_{\mathcal{B}_r}$ are the state coverage of $\mathcal{D}$ and $\mathcal{B}_r$.

**Training Teacher:** An overview of our methodology is depicted in Fig 2. We first construct a trainable actor network $\pi_t^\omega$ parameterized by $\omega$ from $\mathcal{D}$, Fig 2 step 1. For training $\pi_t^\omega$ synthetic data $\hat{S}$ is generated by sampling states from a uniform random distribution over state boundaries $B(s)$, $\hat{S} = \mathcal{U}(B(S))$. Note that this does not represent the true state distribution and may have state combinations that will never occur. We train $\pi_t^\omega$ using behavior cloning where state $\hat{s} \sim \hat{S}$ is checked with root decision node in Eq. 2. A random action is chosen if $\hat{s}$ does not satisfy decision node $T_{\eta_0}$ or leaf action is absent. If $\hat{s}$ satisfies a $T_{\eta_i}$, $T_{\eta_i}$ is traversed and action $a_{\eta_i}$ is returned from the leaf node. This is illustrated in Fig 2 (a). We term the pre-trained actor network $\pi_t^\omega$ as the teacher policy.

**Regularizing Critic:** We now introduce Algo 1 (App C) to train an offline RL agent on $\mathcal{B}_r$. Algo 1 takes $\mathcal{B}_r$ and pretrained $\pi_t^\omega$ as input. The algorithm uses two hyper-parameters, warm start parameter $k$ and mixing parameter $\lambda$. A critic network $Q_s^\theta$ with Monte-Carlo (MC) dropout and target network $Q_s^{\theta'}$ are initialized. ExID is divided into two phases. In the first phase, we aim to warm start the critic network $Q_s^\theta$ with actions from $\pi_t^\omega$ as shown in Fig 2b( i). However, this must be done selectively as the teacher's policy is random around the states that do not satisfy domain knowledge. In each iteration, we first check the states sampled from a mini-batch of $\mathcal{B}_r$ with $\mathcal{D}$. For the states which satisfy $\mathcal{D}$ we compute the teacher action $\pi_t^\omega(s)$ and critic's action $\text{argmax}_a(Q_s^\theta(s, a))$ and collect it in lists $a_t, a_s$, Algo 1 lines 4-10. Our main objective is to keep actions chosen by the critic network

for $s \models \mathcal{D}$ close to the teacher's policy. To achieve this, we introduce a regularization term:

$$\mathcal{L}_r(\theta) = \underbrace{\mathbb{E}_{s \sim \mathcal{B}_r \wedge s \models \mathcal{D}}}_{\text{states matching domain rule}} \underbrace{[Q_s^\theta(s, a_s) - Q_s^\theta(s, a_t)]^2}_{\text{Q regularizer}} \tag{3}$$

Eq 3 incentivizes the critic to increase Q values for actions from $\pi_t^\omega$ and decreases Q values for other actions when $\text{argmax}_a(Q_s^\theta(s, a)) \neq \pi_t^\omega(s)$ for states that satisfy domain knowledge. Note that Eq 3 will only be 0 when $\text{argmax}_a(Q_s^\theta(s, a)) = \pi_t^\omega(s)$ for $s \models \mathcal{D}$. It is also set to 0 for $s \not\models \mathcal{D}$. However, since $\pi_t^\omega$ mimicking heuristic rules is sub-optimal, it is also important to incorporate learning from the data. The final loss is a combination of Eq. 1 and Eq. 3 with a mixing parameter $\lambda \in [0, 1]$:

$$\mathcal{L}(\theta) = \mathcal{L}_{cql}(\theta) + \lambda \mathbb{E}_{s \sim \mathcal{B}_r \wedge s \models \mathcal{D}}[Q_s^\theta(s, a_s) - Q_s^\theta(s, a_t)]^2 \tag{4}$$

The choice of $\lambda$ and the warm start parameter $k$ depends on the quality of $\mathcal{D}$. In the case of perfect domain knowledge, $\lambda$ would be set to 1, and setting $\lambda$ to 0 would lead to the vanilla CQL loss. Mixing both the losses allows the critic to learn both from the data in $\mathcal{B}_r$ and knowledge in $\mathcal{D}$.

**Updating Teacher:** Given a reasonable warm start, the critic is expected to give higher Q values for optimal actions for $s \in \mathcal{D} \cap \mathcal{B}_r$ as it learns from data. We aim to leverage this knowledge to enhance the initial teacher policy $\pi_t^\omega$ trained on heuristic domain knowledge. For $s \sim \mathcal{B}$ and $s \models \mathcal{D}$, we calculate the average Q-values over actions suggested by the critic and the teacher, and compare them as outlined in Algo 1 line 11 referring to Cond. 6. For brevity $\mathbb{E}_{s \sim \mathcal{B}_r \wedge s \models \mathcal{D}}$ is written as $\mathbb{E}$.

If $\mathbb{E}(Q_s^\theta(s, a_s)) > \mathbb{E}(Q_s^\theta(s, a_t))$ , this indicates that the critic expects a higher average return from its action than from the teacher's action. In such cases, we can use the critic's action to update $\pi_t^\omega$, thereby improving the teacher policy over the domain $\mathcal{D}$. However, solely relying on the critic's Q-values can be misleading, as high Q-values may appear for out-of-distribution (OOD) actions. To prevent the teacher from being updated by OOD actions, we measure the average uncertainty of the Q-values for both the critic and teacher actions.

Uncertainty has been shown to be a good metric for OOD action detection by (Wu et al., 2021; An et al., 2021). A well-established methodology to capture uncertainty is predictive variance, which takes inspiration from Bayesian formulation for the critic function and aims to maximize $p(\theta|X, Y) = p(Y|X, \theta)p(\theta)/p(Y|X)$, where $X = (s, a)$ and $Y$ represents the true Q value of the states. However, $p(Y|X)$ is generally intractable we approximate it using Monte Carlo (MC) dropout, which involves including dropout before every layer of the critic network and using it during inference (Gal & Ghahramani, 2016).

Following (Wu et al., 2021), we measure the uncertainty of prediction using Eq 5.

$$Var^T[Q(s, a)] \approx \frac{1}{T} \sum_{t=1}^{T} [Q(s, a) - \bar{Q}(s, a)]^2 \tag{5}$$

Eq 5 estimates the variance of Q value $Q(s, a)$ for an action $a$ using $T$ forward passes on the $Q_s^\theta(s, a)$ with dropout where $\bar{Q}(s, a)$ represents the predictive mean. We then check the average uncertainty of Q-values for actions chosen by the critic and teacher policies over states in the batch that match the domain knowledge. The teacher network is updated using the critic's action only if the critic's policy has a higher expected Q-value than the teacher's and the uncertainty of this action is lower than that of the teacher's action. If $\mathbb{E}(Var^T Q_s^\theta(s_r, a_s)) < \mathbb{E}(Var^T Q_s^\theta(s_r, a_t))$ , it suggests that the critic's actions are learned from expert data in the buffer and are not OOD samples. The condition is summarized in cond. 6:

$$\mathbb{E}(Q_s^\theta(s_r, a_s)) > \mathbb{E}(Q_s^\theta(s_r, a_t)) \wedge$$
$$\mathbb{E}(Var^T Q_s^\theta(s_r, a_s)) < \mathbb{E}(Var^T Q_s^\theta(s_r, a_t)) \tag{6}$$

We update the teacher with cross-entropy described in Eq 7:

$$\mathcal{L}(\omega) = - \sum_{s \models D} (\pi_t^\omega(s) log(\pi_s(s))) \tag{7}$$

where, $\pi_s(s, a) = \frac{e^{Q(s,a)}}{\sum_{a'} Q(s,a')}$. When the critic's policy is better than the teacher's policy, $\mathcal{L}_r(\theta)$ is set to 0 Algo 1 Lines 11 to 13. Finally, the critic network is updated using calculated loss $\mathcal{L}(\theta)$ Algo 1 Lines 17-18. We study the theoretical implications of using domain knowledge based regularization with simplified assumptions in App. A.

Furthermore, we extend this to continuous domain by using the regularization in Eq 4 during critic $(Q_s^\theta)$ training for continuous domain and using actions from actor network $(\pi_s)$ for cross entropy loss in Eq 7.

## 5 EMPIRICAL EVALUATIONS

We investigate the following through our empirical evaluations: *1. Does ExID perform better than combining $\mathcal{D}$ and offline RL algos on different environments with datasets exhibiting rare and OOD states Sec 5.2? 2. Does ExID generalize to OOD states covered by $\mathcal{D}$ Sec 5.4? 3. What is the effect of varying $k$, $\lambda$ and updating $\pi_t^\omega$ Sec 5.5? 4. How does performance vary with the quality of $\mathcal{D}$ Sec 5.6?*

### 5.1 EXPERIMENTAL SETTING

We evaluate our methodology on open-AI gym (Brockman et al., 2016), MiniGrid (Chevalier-Boisvert et al., 2023), *real sales promotion (SP) (Qin et al., 2022)* and sim-glucose (Gao, 2024) offline data sets. All our data sets are generated using standard methodologies defined in (Schweighofer et al., 2022; 2021) *except SP which is generated by human operators.* All experiments have been conducted on a Ubuntu 22.04.2 LTS system with 1 NVIDIA K80 GPU, 4 CPUs, and 61GiB RAM. App. G notes the hyperparameter values and network architectures.

**Dataset:** We experiment on three types of data sets. *Expert Data-set* (Fu et al., 2020; Gulcehre et al., 2021; Kumar et al., 2020) generated using an optimal policy without any exploration with high trajectory quality but low state action coverage. *Replay Data-set* (Agarwal et al., 2020; Fujimoto et al., 2019b) generated from a policy while training it online, exhibiting a mixture of multiple behavioral policies with high trajectory quality and state action coverage. *Noisy Data-set* (Fujimoto et al., 2019a;b; Kumar et al., 2020; Gulcehre et al., 2021) generated using an optimal policy that also selects random actions with $\epsilon$ greedy strategy where $\epsilon = 0.2$ having low trajectory quality and high state action coverage. Additionally we also experiment on human generated dataset for sales promotion task and simglucose task.

**Baselines:** We do comparative studies on 10 baselines for OpenAI gym datasets. The first baseline simply checks the conditions of $\mathcal{D}$ and applies corresponding actions in execution. The performance of this baseline shows that $\mathcal{D}$ is imperfect and does not achieve the optimal reward. CQL SE is from (Verma et al., 2024) where the expert is replaced by $\mathcal{D}$. The other baselines are an ensemble of $\mathcal{D}$ and eight algorithms popular in the Offline RL literature for discrete environments. These algorithms include Behavior Cloning (BC) (Pomerleau, 1991), Behaviour Value Estimation (BVE) (Gulcehre et al., 2021), Quantile Regression DQN (QRDQN) (Dabney et al., 2018), REM, MCE, BCQ, CQL and Critic Regularized Regression Q-Learning (CRR) (Wang et al., 2020). *For a fair comparison, we use actions from domain knowledge for states not in the buffer and actions from the trained policy for other states to obtain the final reward.* Hence, each algorithm is renamed with the suffix D in Table 5.1.

**Limiting Data:** To create limited-data settings for benchmark datasets, we first extract a small percentage of samples from the full dataset and remove some of the samples based on state conditions. This is done to ensure the reduced buffer satisfies the conditions defined in Def 4.1. We describe the specific conditions of removal in the next section. Further insights and the state visualizations for selected reduced datasets are in App I. **Note : no data reduction has been performed on SP dataset to demonstrate a real dataset exhibits characteristics of reduced buffer**.

### 5.2 PERFORMANCE ACROSS DIFFERENT DATASETS

Our results for OpenAI gym environments are summarized in Table 5.1 and Minigrid in Table 4 (App E). We observe the performance of offline RL algorithms degrades substantially when part

Table 1: Average reward [↑] obtained during online evaluation over 3 seeds on openAI gym envs

| ENV DATA | DATA TYPE | $\mathcal{D}$ | QRDQN D | REM D | BVE D | CRR D | MCE D | BC D | BCQ D | CQL D | CQL SE | CQL (FULL) | **ExID (OURS)** |
|---|---|---|---|---|---|---|---|---|---|---|---|---|---|
| MOUNTAIN CAR | EXPERT | -159.9 ± 52.28 | -168.2 ± 33.71 | -147.7 ± 21.54 | -175.36 ± 25.16 | -157.2 ± 39.09 | -152 ± 37.41 | -181.38 ± 28.60 | -172.9 ± 27.5 | -167.49 ± 12.3 | -161.33 ± 18.57 | -128.63 ± 10.94 | -125.5 ± 2.60 |
| | REPLAY | | -137.14 ± 39.27 | -136.26 ± 40.15 | -152.0 ± 35.06 | -137.23 ± 42.79 | -139.91 ± 40.01 | -137.26 ± 43.04 | -136.29 ± 36.15 | -140.38 ± 33.58 | -150.67 ± 16.68 | -135.4 ± 3.74 | -105.79 ± 11.38 |
| | NOISY | | -141.61 ± 33.04 | -134.99 ± 32.60 | -173.95 ± 39.60 | -178.99 ± 23.58 | -168.69 ± 38.78 | -140.0 ± 28.5 | -144.52 ± 43.04 | -179.8 ± 29.99 | -126.96 ± 17.84 | -107.06 ± 12.73 | -109.9 ± 13.45 |
| CART POLE | EXPERT | 57.0 ± 5.35 | 33.23 ± 3.17 | 41.31 ± 8.76 | 16.16 ± 9.41 | 15.24 ± 5.62 | 16.1 ± 4.4 | 225.76 ± 74.39 | 165.36 ± 15.01 | 121.8 ± 14.0 | 155.78 ± 26.47 | 364.1 ± 22.15 | 307.18 ± 137.72 |
| | REPLAY | | 149.09 ± 14.05 | 180.70 ± 62.79 | 11.1 ± 2.13 | 11.24 ± 2.71 | 9.16 ± 0.25 | 144.43 ± 2.41 | 144.76 ± 6.04 | 131.97 ± 23.23 | 113.37 ± 5.88 | 250.02 ± 55.02 | 340.26 ± 30.58 |
| | NOISY | | 161 ± 6.40 | 15.33 ± 0.58 | 11.53 ± 3.77 | 13.68 ± 7.49 | 10.66 ± 2.04 | 68.4 ± 14.67 | 63.53 ± 14.08 | 92.6 ± 22.05 | 92.6 ± 22.05 | 93.72 ± 37.79 | 228.61 ± 38.64 |
| LUNAR LANDER | EXPERT | 52.48 ± 26.51 | 5.14 ± 25.10 | -184.84 ± 26.45 | -681.67 ± 34.86 | 8.79 ± 25.38 | 19.71 ± 10.52 | 38.40 ± 23.21 | -45.99 ± 30.47 | 65.43 ± 71.37 | 53.22 ± 78.85 | 167.74 ± 29.4 | 161.34 ± 17.10 |
| | REPLAY | | -444.20 ± 12.20 | -556.81 ± 21.39 | -572 ± 27.93 | -131.21 ± 31.97 | -115.23 ± 18.16 | 136.63 ± 12.40 | 111.47 ± 54.67 | 61.83 ± 45.57 | 87.70 ± 18.20 | 187.72 ± 25.62 | 156.03 ± 56.67 |
| | NOISY | | -4.81 ± 97.28 | 21.41 ± 14.71 | 28.65 ± 12.26 | -158.27 ± 7.71 | -50.47 ± 15.78 | 98.62 ± 28.01 | 101.59 ± 30.83 | 5.01 ± 128.63 | 40.35 ± 65.72 | 111 ± 52.32 | 163.57 ± 49.24 |

of the data is not seen and trajectory ratios change. For these cases with only 10% partial data, ExID surpasses the performance by at least 27% in the presence of reasonable domain knowledge. The proposed method performs strongest on the replay dataset where the contribution of $L_r(\theta)$ is significant due to state coverage, and the teacher learns from high-quality trajectories. Environment details are described in the App. E. All domain knowledge trees are shown in the App. E Fig 10. We describe limiting data conditions and domain knowledge specific to the environment as follows:

**Mountain Car Environment:** (Moore, 1990) We use simple, intuitive domain knowledge in this environment shown in the App. E Fig 10 (c), which represents taking a left action when the car is at the bottom of the valley with low velocity to gain momentum; otherwise, taking the right action to drive the car up. Fig 6 (c) shows the state action pairs this rule generates on states sampled from a uniform random distribution over the state boundaries. It can be observed that the states of $\mathcal{D}$ cover part of the missing data in Fig 1 (a). For limiting datasets, we remove states with position > -0.8. The performance of CQLD and ExID are shown in Fig 3 (a),(b) where ExID surpasses CQLD for all three datasets.

**Cart-pole Environment:** For this environment, we use domain knowledge from (Silva & Gombolay, 2021), which aims to move in the direction opposite to the lean of the pole, keeping the cart close enough to the center. If the cart is close to an edge, the domain knowledge attempts to account for the cart's velocity and recenter the cart. The full tree is given in the App. E Fig 10 (a). We remove states with cart velocity > -1.5 to create the reduced buffer.

**Lunar-Lander Environment:** We borrow the decision nodes from (Silva et al., 2020) and get actions from a sub-optimal policy trained online with an average reward of 52.48. The full set of decision nodes is shown in the App. E Fig 10 (b). $\mathcal{D}$ focuses on keeping the lander balanced when the lander is above ground. When the lander is near the surface, $\mathcal{D}$ focuses on keeping the y velocity lower. To create the reduced datasets, we remove data of lander angle < -0.04.

**Mini-Grid Environments:** For our experiments, we choose two environments: Random Dynamic Obstacles 6X6 and LavaGapS 7X7. We use intuitive domain knowledge which avoids crashing into obstacles in front, left, or right of agent ref. App. E Fig 10 (d), (e). We remove states with obstacles on the right for creating limited data settings. Due to limitation of space we report the results of the best-performing algorithms on the replay dataset in Table 4 (App E).

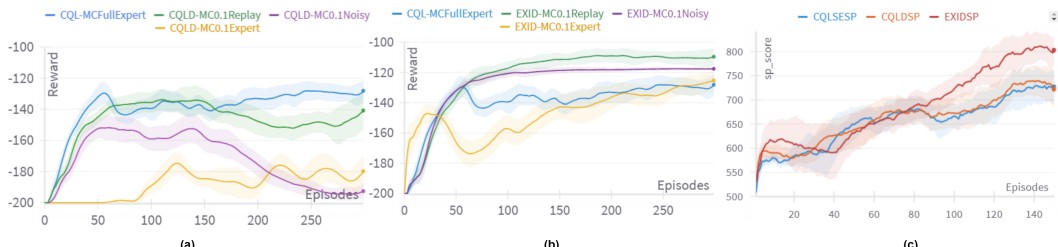

Figure 3: Performance of (a) CQL and (b) EXID on all datasets for Mountain Car during online evaluation (c) Evaluation curves for the sales promotion dataset

### 5.3 CASE STUDY ON REAL HUMAN GENERATED SALES PROMOTION (SP) AND SIM-GLUCOSE DATASET

SP dataset and environment (Qin et al., 2022) simulates a real-world sales promotion platform. The number of coupons and the discount the user received will affect his behavior. A higher discount will promote the sales, but the cost will also increase. The goal for the platform operator is to maximize the total profit. The horizon of the dataset is 50 days for the training and 30 days for the test. Domain knowledge ((Qin et al., 2022), App A] : Active users can be given more coupons with lower discount to maximize profit. We model this as $order_{number} > 60 \wedge Avg_{fee} > 0.8 \implies [5, 0.95]$ where action 1 is number of coupons range [0,5] and action 2 is coupon value (discount value = (1-coupon value)) range [0.6,0.95]. The dataset exhibits the properties in Def 4.1 as first 50 days of sales does not contain many active users (20.32%) depicting scarcity. The domain rule is imperfect as coupon value and number depend on multiple factors such as user purchase history and behavior. As illustrated in the table 2 and Fig 3 (c) the intuitive domain rule enhances performance by 10.49% in the real dataset. Comparison with other popular offline RL baselines are provided in App D. The simglucose (Gao, 2024) dataset is obtained from Type 1 diabetes simulation with domain knowledge: 1. The basal insulin is based on the insulin amount to keep the blood glucose in the steady state when there is no (meal) disturbance.

$$(meal = 0) \implies \text{basal} = u2ss \,(\text{pmol/L*kg}) \times \text{body\_weight (kg)}/6000 \,(\text{U/min})$$

2. The bolus amount is computed based on the current glucose level, the target glucose level, the patient's correction factor, and the patient's carbohydrate ratio.

$$(meal > 0) \implies \text{bolus} = \left(\frac{\text{carbohydrate}}{\text{carbohydrate\_ratio}}\right) + \left(\frac{\text{current\_glucose} - \text{target\_glucose}}{\text{correction\_factor}}\right)/\text{time}$$

Table 2: Results on human-generated Sales Promotion and SimGlucose datasets

| Dataset | $\mathcal{D}$ | CQL + $\mathcal{D}$ | CQLSE | EXID | MOPO |
|---|---|---|---|---|---|
| Sales Promotion | $654.68 \pm 20.06$ | $722.06 \pm 71.40$ | $727.03 \pm 49.56$ | $802.91 \pm 41.69$ | $404.48 \pm 7.39$ |
| Sim Glucose | $17.53 \pm 3.02$ | $21.79 \pm 3.60$ | $24.28 \pm 2.45$ | $30.82 \pm 3.95$ | $34.64 \pm 28.13$ |

### 5.4 GENERALIZATION TO OOD STATES AND CONTRIBUTION OF $\mathcal{L}_r(\theta)$

In Fig 4 (a), (b), we plot $Q_s^\theta(s, a_{expert}) - Q_s^\theta(s, a_\theta)$ for CQL and EXID policies for different datasets of Mountain-Car environments. Action $a_{expert}$ is obtained from the full expert dataset where position $> -0.8$. We observe that the Q value for actions of CQL policy diverges from the expert policy actions with high values for the states not in the reduced buffer, whereas ExID stays close to the expert actions for the unseen states. This empirically shows generalization to OOD states not in the dataset but covered by domain knowledge. In Fig 4 (d), we plot the contribution by $\mathcal{L}_r(\theta)$ during the training and observe the contribution is higher for replay data sets with more state coverage.

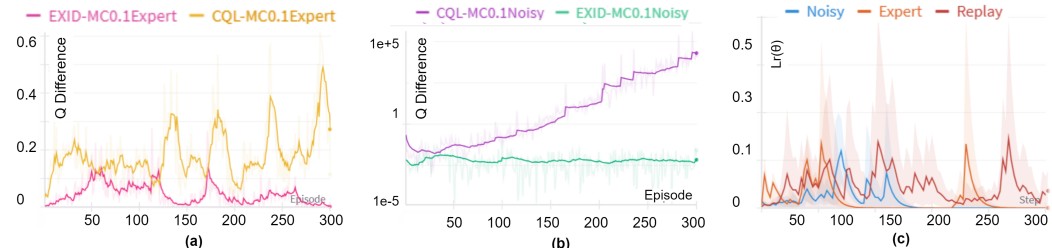

Figure 4: Q value difference between CQL and EXID for expert and policy action on states not present in the buffer for a) expert b) noisy in log scale c) contribution of $\mathcal{L}_r(\theta)$

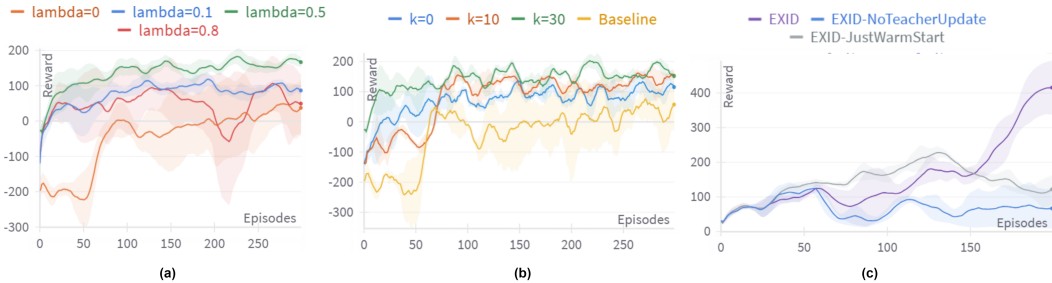

Figure 5: (a) Effect of different $\lambda$ on the performance of ExID on Lunar Lander (b) Effect of different $k$ on the performance of EXID on Lunar Lander (c) Performance of EXID with teacher update, no teacher update, and just warm start on Cart-pole.

## 5.5 PERFORMANCE ON VARYING $\lambda$, $k$, AND ABLATION OF $\pi_t^\omega$

We study the effect of varying $\lambda$ on the algorithm for the given domain knowledge. We empirically observe setting a high or a low $\lambda$ can yield sub-optimal performance, and $\lambda = 0.5$ generally yields good performance. In Fig 5 (a), we show this effect for LunarLander. Plots for other environments are in the App. H Fig 11. For $k$ we observe setting the warm start parameter to 0 yields a sub-optimal policy, as the critic may update $\pi_t^\omega$ without completely learning from it. The starting performance increases with an increase in $k$ as shown in Fig 5 (b) for LunarLander. $k = 30$ works best according to empirical evaluations. Plots for other environments are in the App. H Fig 12. We show two ablations for Cart-pole in Fig 5 (c) with no teacher update after the warm start and no inclusion of $\mathcal{L}_r(\theta)$ after the warm start. The warm start in this environment is set to 30 episodes. Fig 5 c) shows without teacher updated, the sub-optimal teacher drags down the performance of the policy beyond the warm start, exhibiting the necessity of $\pi_t^\omega$ update. Also, the student converges to a sub-optimal policy if no $\mathcal{L}_r(\theta)$ is included beyond the warm start.

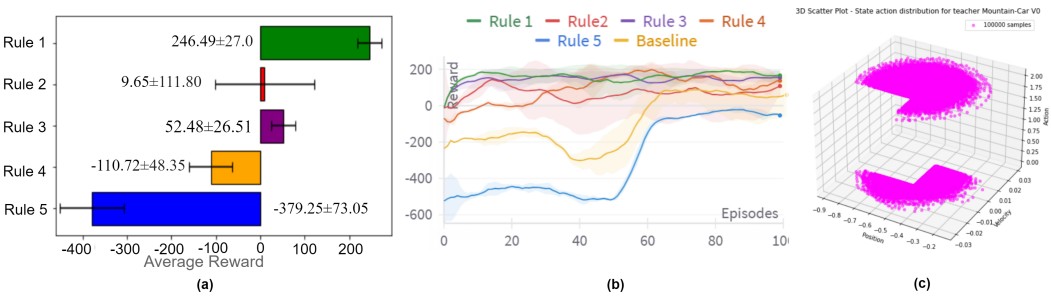

Figure 6: (a) $\mathcal{D}$ with different average rewards (b) Performance effect on Lunar-lander (c) State distribution generated for training the teacher network for mountain-car

## 5.6 EFFECT OF VARYING $\mathcal{D}$ QUALITY

We show the effect of choosing policies as $\mathcal{D}$ with different average rewards for Lunar-Lander expert data in Fig 6 (a) and (b). Rule 1 is optimal and has almost the same effect as Rule 3, which is the $\mathcal{D}$ used in our experiments exhibiting that updating a sub-optimal $\mathcal{D}$ can lead to equivalent performance as optimal $\mathcal{D}$. Using a rule with high uncertainty, as Rule 2, induces high uncertainty in the learned policy but performs slightly better than the baseline. Rule 4, which has a lower average reward, also causes gains on average performance with slower convergence. Finally, Rule 5, with very bad actions, affects policy performance adversely and leads to a performance lower than baseline CQL.

## 6 CONCLUSION AND LIMITATION

In this paper, we study the effect of limited and partial data on offline RL and observe that the performance of SOTA offline RL algorithms is sub-optimal in such settings. The paper proposes a methodology to handle offline RL's performance degradation using domain insights. We incorporate a regularization loss in the CQL training using a teacher policy and refine the initial teacher policy while training. We show that incorporating reasonable domain knowledge in offline RL enhances performance, achieving a performance close to full data. However, this method is limited by the quality of the domain knowledge and the overlap between domain knowledge states and reduced buffer data. In the future, the authors would like to improve on capturing domain knowledge into the policy network without dependence on data and enhancing the method to work with more general forms of domain knowledge.

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

## A   THEORETICAL IMPLICATIONS WITH SIMPLIFIED ASSUMPTIONS

**Notations**

For any deterministic policy $\pi$ the performance return is formulated as $\eta(\pi) = \mathbb{E}_{\tau \sim \pi}[\sum_{t=0}^{\infty} \gamma^t r(s_t, a_t)]$

For any policy $\pi$, $\rho_\pi$ is the (unnormalized) discounted visitation frequency given by $\rho_\pi(s) = \sum_{t=0}^{\infty} \gamma^t P(s_t = s)$ where $s_0 \sim \rho^0(s_0)$ and the trajectory $(s_0, s_1, \dots)$ is sampled from the policy $\pi$ and $\rho_\pi(s) \in [0, \frac{1}{1-\gamma}]$. $\bar{\rho}_\pi(s) = sup\{\rho_\pi(s), s \in S\} \in [\frac{1}{|S_\pi|(1-\gamma)}, \frac{1}{(1-\gamma)}]$

We denote the regularized policy learned by ExID on $\mathcal{B}_r$ as $\hat{\pi}$ and the unregularized policy as $\pi_u$.

**Lemmas**

We introduce the following Lemma required for our theoretical analysis.

**Lemma A.1.** *((Yang et al., 2023)) Given two policies $\pi_1$ and $\pi_2$*

$$\eta(\pi_1) - \eta(\pi_2) = \int_{s \in S} \rho_{\pi_1}(s)(Q^*(s, \pi_1(s) - V^*(s))ds - \int_{s \in S} \rho_{\pi_2}(s)(Q^*(s, \pi_2(s) - V^*(s))ds$$

*Proof.* Please refer to Lemma A.1 Eq 17 in (Yang et al., 2023)                    □

**Proposition A.2.** *Denote $\hat{\pi}$ as the policy learned by ExID, $\pi_u$ as any offline RL policy learned on $\mathcal{B}_r$ and optimal Q function as $Q^*$ and V function as $V^*$. Then it holds that*

$$\eta(\hat{\pi}) - \eta(\pi_u) \geq \mathbb{E}_{s \sim O|\pi_u}[V^*(s) - Q^*(s, \pi_u(s))] - \bar{\rho}_{\hat{\pi}}\alpha$$

*Proof.* According to (Kakade & Langford, 2002) performance improvement between two policies if given by

$$\eta(\pi_1) = \eta(\pi_2) + \mathbb{E}_{\tau \sim \pi_1}\left[\sum_{t=0}^{\infty} \gamma^t Q_{\pi_2}(s_t, a_t) - V_{\pi_2}(s_t)\right] \tag{8}$$

Replacing $\pi_1$ by $\hat{\pi}$ and $\pi_2$ by $\pi_u$ and by following Lemma A.1

$$\eta(\hat{\pi}) - \eta(\pi_u) = \int_{s \in S} \rho_{\hat{\pi}}(s)(Q^*(s, \hat{\pi}(s)) - V^*(s))ds - \int_{s \in S} \rho_{\pi_u}(s)(Q^*(s, \pi_u(s)) - V^*(s))ds \tag{9}$$

$$= \int_{s \in S} \rho_{\pi_u}(s)(V^*(s) - Q^*(s, \pi_u(s)))ds - \int_{s \in S} \rho_{\hat{\pi}}(s)(V^*(s) - Q^*(s, \hat{\pi}(s)))ds \tag{10}$$

Dividing the state space into in dataset domain states (I) and OOD states (O). The

$$\tag{11}$$

$$\underbrace{\left[\int_{s \in I} \rho_{\pi_u}(s)(V^*(s) - Q^*(s, \pi_u(s)))ds - \int_{s \in I} \rho_{\hat{\pi}}(s)(V^*(s) - Q^*(s, \hat{\pi}(s)))ds\right]}_{a} +$$

$$\underbrace{\left[\int_{s \in O} \rho_{\pi_u}(s)(V^*(s) - Q^*(s, \pi_u(s)))ds - \int_{s \in O} \rho_{\hat{\pi}}(s)(V^*(s) - Q^*(s, \hat{\pi}(s)))ds\right]}_{b} \tag{12}$$

Since the regularization loss facilitates visitation to OOD states via knowledge distillation we assume $\rho_{\hat{\pi}} = \rho_{\pi_u} - \Delta_i$ for $s \in i$ and $\rho_{\hat{\pi}} = \rho_{\pi_u} + \Delta_o$ for $s \in o$ where $\Delta_i \in [0, \rho_{\pi_u(s)}]$ and $\Delta_o \in [0, \frac{1}{1-\gamma} - \rho_{\pi_u(s)}]$

$$a = \int_{s \in I} \rho_{\pi_u}(s)(V^*(s) - Q^*(s, \pi_u(s)))ds - \int_{s \in I} (\rho_{\pi_u} - \Delta_i)(s)(V^*(s) - Q^*(s, \hat{\pi}(s)))ds \quad (13)$$

$$= \int_{s \in I} \rho_{\pi_u}(s)(Q^*(s, \hat{\pi}(s)) - Q^*(s, \pi_u(s)))ds + \int_{s \in I} \Delta_i(s)(V^*(s) - Q^*(s, \hat{\pi}(s)))ds \quad (14)$$

Under assumption in distribution action can be learned from the dataset due to conservatism of offline RL $(Q^*(s, \hat{\pi}(s)) - Q^*(s, \pi_u(s))) \approx 0$, $a \geq 0$

$$b = \int_{s \in O} \rho_{\pi_u}(s)(V^*(s) - Q^*(s, \pi_u(s)))ds - \int_{s \in O} (\rho_{\pi_u} + \Delta_o)(s)(V^*(s) - Q^*(s, \hat{\pi}(s)))ds \quad (15)$$

$$\geq \int_{s \in O} \rho_{\pi_u}(s)(V^*(s) - Q^*(s, \pi_u(s)))ds - \int_{s \in O} \rho_{\hat{\pi}}(s)(V^*(s) - Q^*(s, \hat{\pi}(s)))ds \quad (16)$$

$$\geq \mathbb{E}_{s \sim O|\pi_u}[V^*(s) - Q^*(s, \pi_u(s))] - \mathbb{E}_{s \sim O|\hat{\pi}}[V^*(s) - Q^*(s, \hat{\pi}(s))] \quad (17)$$

Further loosening the lower bound

$$= \mathbb{E}_{s \sim O|\pi_u}[V^*(s) - Q^*(s, \pi_u(s))] - \bar{\rho}_{\hat{\pi}} \int_{s \in O} \frac{\rho_{\hat{\pi}}}{\bar{\rho}_{\hat{\pi}}}(V^*(s) - Q^*(s, \hat{\pi}(s)))ds \quad (18)$$

$$\geq \mathbb{E}_{s \sim O|\pi_u}[V^*(s) - Q^*(s, \pi_u(s))] - \bar{\rho}_{\hat{\pi}} \int_{s \in O} (V^*(s) - Q^*(s, \hat{\pi}(s)))ds \quad (19)$$

Combining Eq 14, 17 and 19, and denoting $\alpha = \mathbb{E}_{s \sim O}[V^*(s) - Q^*(s, \hat{\pi}(s))]$

$$\eta(\hat{\pi}) - \eta(\pi_u) \geq \mathbb{E}_{s \sim O|\pi_u}[V^*(s) - Q^*(s, \pi_u(s))] - \bar{\rho}_{\hat{\pi}}\alpha \quad (20)$$

Hence, Proposition A.2 follows **Q.E.D**

$\square$

## B Missing Examples

*Performing $Q - Learning$ by sampling from a reduced batch $\mathcal{B}_r$ may not converge to an optimal policy for the MDP $M_\mathcal{B}$ representing the full buffer.*

**Example** (Theorem 1,(Fujimoto et al., 2019b)) defines MDP $M_\mathcal{B}$ of $\mathcal{B}$ from same state action space of the original MDP $M$ with transition probabilities $p_\mathcal{B}(s'|s,a) = \frac{N(s,a,s')}{\sum_{\tilde{s}} N(s,a,\tilde{s})}$ where $N(s,a,s')$ is the number of times $(s,a,s')$ occurs in $\mathcal{B}$ and an terminal state $s_{init}$. It states $p_\mathcal{B}(s_{init}|s,a) = 1$ when $\sum_{\tilde{s}} N(s,a,\tilde{s}) = 0$. This happens when transitions of some $s'$ of $(s,a,s')$ are missing from the buffer, which may occur in $\mathcal{B}_r$ when $\mathcal{B}_r \subset \mathcal{B}$. $r(s_{init}, s, a)$ is initialized to $Q(s,a)$. We assume that a policy learned on reduced dataset $\mathcal{B}_r$ converges to optimal value function and disprove it using the following counterexample:

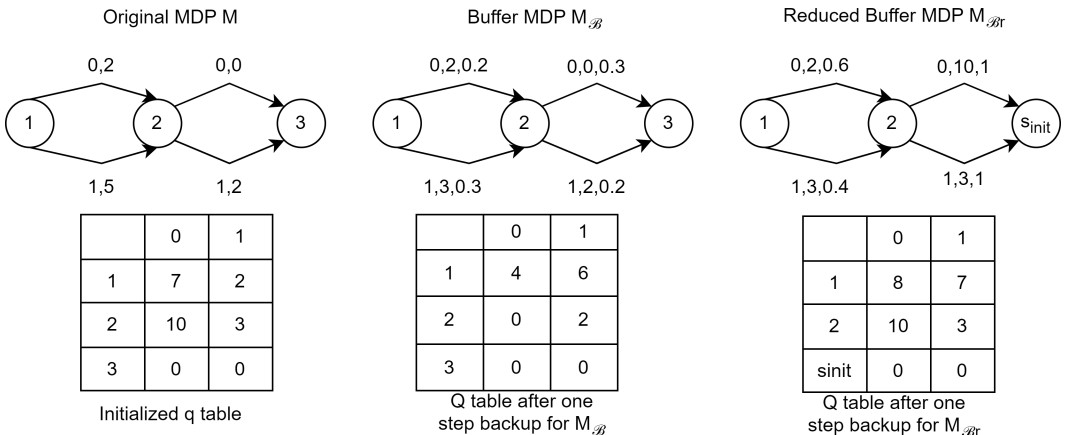

Figure 7: Example MDP, sampled buffer MDP and reduced buffer with Q tables

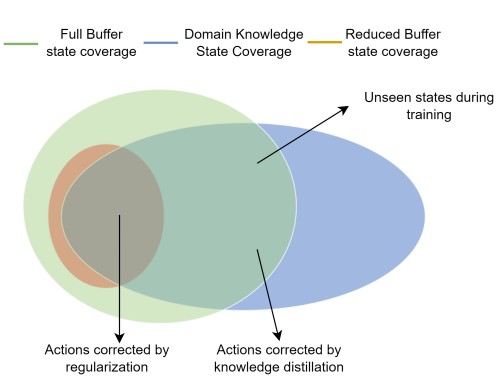

Figure 8: We hypothesize the suboptimal performance of offline RL for limited data can be addressed via domain knowledge via action regularization and knowledge distillation.

A visualization is shown in Fig 8.

We take a simple MDP illustrated in Fig 7 with 3 states and 2 actions (0,1). The reward of each action is marked along the transition. The sampled MDP is constructed the following samples (1,0,2)-2,(1,1,2)-3, (2,0,3)-3, and (2,1,3)-2 and the reduced buffer MDP with samples (1,0,2)-2 and (1,1,2)-1. The probabilities are marked along the transition. It is easy to see that the policy learned under the reduced MDP converges to a nonoptimal policy after one step of the Q table update with $Q(s,a) = r(s,a) + p(s'|s,a) * max_{a'}(Q(s',a'))$. This happens because of transition probability shift on reducing samples $p_\mathcal{B}(s'|s,a) \neq p_{\mathcal{B}_r}(s'|s,a)$ and no Q updates for $(s,a) \notin \mathcal{B}_r$.

Our methodology addresses these issues as follows:

- For $s \in D \cap \mathcal{B}_r$ better actions are enforced through regularization using $\pi_t^\omega$ even when the transition probabilities are low for optimal transitions.

- Incorporating regularization distills the teacher's knowledge in the critic-enhancing generalization.

## C  ALGORITHM

The pseudo code of the algorithm is described in Algo 1.

---

**Algorithm 1** Pseudo code for EXID

---

1: **Input:** Reduced buffer $\mathcal{B}_r$, Initial teacher network $\pi_t^\omega$, Training steps $N$, Warm-up steps $k$, Soft update $\tau$, hyperparameters: $\lambda, \alpha$
2: Initialize Critic with MC dropout and Target Critic $Q_s^\theta, Q_s^{\theta'}$
3: **for** $n \leftarrow 1$ **to** $N$ **do**
4:     Sample mini-batch $b$ of transitions $(s, a, r, s') \sim \mathcal{B}_r$ $a_t = [], a_s = [], s_r = []$
5:     **for** $s \in b$ **do**
6:         **if** $s \models \mathcal{D}$ and $\pi_t^\omega(s) \neq argmax_a(Q_s^\theta(s, a))$ **then**
7:             $a_t.append(\pi_t^\omega(s))$
8:             $a_s.append(\mathrm{argmax}_a(Q_s^\theta(s, a)))$
9:             $s_r.append(s)$
10:        **end if**
11:    **end for**
12:    **if** $n > k \wedge$ Cond. 6 **then**
13:        Update $\pi_t^\omega(s)$ using Eq 7
14:        $\mathcal{L}_r(\theta) = 0$
15:    **else**
16:        Calculate $\mathcal{L}_r(\theta)$ using Eq 3
17:    **end if**
18:    Calculate $\mathcal{L}(\theta)$ using Eq 4
19:    Update $Q_s^\theta$ with $\mathcal{L}(\theta)$ and softy update $Q_s^{\theta'}$ and $\tau$
20: **end for**

---

## D  COMPARISON WITH ADDITIONAL CONTINUOUS DOMAIN BASELINES FOR SP TASK

In this section we compare with additional popular continuous domain baselines. Our experiments show popular offline RL algorithms suffer from generalization to OOD states a problem that can be alleviated by inclusion of reasonable domain knowledge. The baselines are:

**Strategically Conservative Q-Learning (SCQ) (Shimizu et al., 2024):**  SCQ uses a Conditional Variational Autoencoder network to distinguish between OOD actions that are easy or hard to estimate and penalizing the Q values accordingly resulting in a less conservative estimate of action values.

**Adaptive Advantage-Guided Policy Regularization for Offline Reinforcement Learning (Liu et al., 2024):**  A2PR trains a VAE similar to SCQ to identify high advantage that differ from those present in the dataset. The VAE is trained with $\log p\psi(a|s) \geq \mathbb{E}q_\phi(z|a,s)\left[\mathbb{1}f(A(s,a)) > \epsilon_A \log p_\psi(a|z,s)\right] - \mathrm{KL}\left[q\phi(z|a,s) \,\|\, p(z|s)\right]$ where $s \in \mathcal{B}r$. This method does not estimate actions for $s \notin Br$ which ExID does via knowledge distillation.

**Constrained policy optimization with explicit behavior density for offline reinforcement learning (Zhang et al., 2024):** CPED uses a flow-GAN model to explicitly estimate the density of behavior policy. This facilitates choosing different actions which are safe for the for states in dataset. The flow GAN model is trained on the dataset generated by behavior policy and does not account for the states outside the dataset.

**MOPO: Model-based Offline Policy Optimization (Yu et al., 2020):** Model based RL methods in the offline RL setting have been proven to perform better as they aim at learning the model dynamics from the data. These methods then learn on a MDP based on the dynamics with the reward function penalized by an estimate of the model's error. While these methods have outperformed model free methods in settings where the underlying dynamics is viable to learn from data, our experiments show learning true dynamics under limited data is a harder task. Limited or biased data can lead to errors in learnt model dynamics. Since the performance of these class of algorithms depend on the learnt dynamics MOPO suboptimal in the Sales Promotion task. However, we observe MOPO outperforms EXID in the sim-glucose task. We conjecture this is because glucose-insulin

dynamics often involve smooth and predictable transitions which MOPO can leverage effectively. However, our method is primarily designed to address the generalization gap of offline **model free** RL methods.

In summary all these methods except model based RL methods (which depend on the learned dynamics) do not employ any action correction mechanism for OOD states outside the dataset leading to performance degradation in case of limited data. As a result these algorithms perform almost similarly on sales promotion dataset. ExID distills knowledge for OOD states from domain knowledge leading to performance enhancement over the baseline methods. The results are summarized in table.

| Environment | SCQ | A2PR | CPED | EXID |
|---|---|---|---|---|
| SP | $708.44 \pm 52.19$ | $712 \pm 32.09$ | $715 \pm 47.31$ | $827.76 \pm 43.79$ |

Table 3: Performance comparison with other continuous domain baselines in the SP environment.

# E    ENVIRONMENTS AND DOMAIN KNOWLEDGE TREES

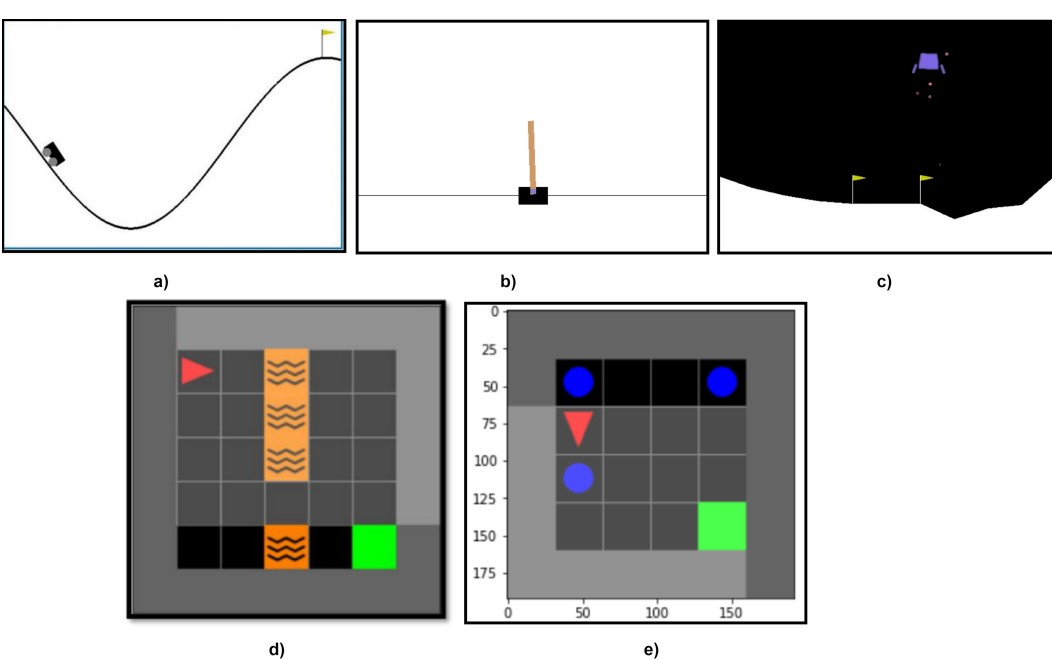

Figure 9: Graphical visualizations of environments used in the experiments. These environments are a) MountainCar-v0 b) CartPole-v1 c) LunarLander-v2 d) MiniGrid-LavaGapS7-v0 e) MiniGrid-Dynamic-Obstacles-Random-6x6-v0

The graphical visualization of each environment is depicted in Fig 9. The choice of environment in this paper depended on two factors: a) Pre-existing standard methods of generating offline RL datasets. b) Possibility of creating intuitive decision tree-based domain knowledge. All datasets have been created via (Schweighofer et al., 2021). We explain the environments in detail as follows:

**Mountain-car Environment:** This environment Fig 9 a) has two state variables, position and velocity, and three discrete actions: left push, right push, and no action (Moore, 1990). The goal is to drive a car up a valley to reach the flag. This environment is challenging for offline RL because of sparse rewards, which are only obtained on reaching the flag.

**Cart-pole Environment** The environment Fig 9 b) has 4 states and 2 actions representing left force and right force. The objective is to balance a pole on a moving cart.

**Lunar-Lander Environment:** The task is to land a lunar rover between two flags Fig 9 c) by observing 8 states and applying one of 4 actions.

**Minigrid Environments:** Mini-grid (Chevalier-Boisvert et al., 2023) is an environment suite containing 2D grid-worlds with goal oriented tasks. As explained in the main text, we experiment using MiniGrid-LavaGapS7-v0 and MiniGrid-Dynamic-Obstacles-Random-6x6-v0 from this environment suite is shown in Fig 9 d) and e). In MiniGrid-LavaGapS7-v0, the agent has to avoid Lava and pass through the gap to reach the goal. Dynamic obstacles are similar; however, the agent can start at a random position and has to avoid dynamically moving balls to reach the goal. The environment has image observation with 3 channels (OBJECT_ID, COLOR_ID, STATE). Following (Schweighofer et al., 2021) experiments, we flatten the image to an array of 98 observations and restrict action space to three actions: Turn left, Turn Right, and Move forward. The results of minigrid environment are reported in Table 4. Since this environment uses a semantic map from image observation, we collect states from a fixed policy with random actions to generate the teacher's state distribution. CQL on the full dataset achieves the average reward of $0.92\pm0.1$ for DynamicObstacles and $0.53 \pm 0.01$ for LavaGapS.

The domain knowledge trees for all the environments are shown in Fig 10. The cart pole domain knowledge tree Fig 10 a) is taken from (Silva & Gombolay, 2021) (Fig 7). The Lunar Lander decision nodes Fig 10 b) have been taken from (Silva et al., 2020) (Fig4). For the mini-grid environments, we construct intuitive decision trees shown in Fig 10 d) and Fig 10 e). Positions 52, 40, and 68 represent positions front, right, and left of the agent. Value 0.2 represents a wall, 0.9 represents Lava, and 0.6 represents a ball. We check positions 52, 40, and 68 for these obstacles and choose the recommended actions as domain knowledge.

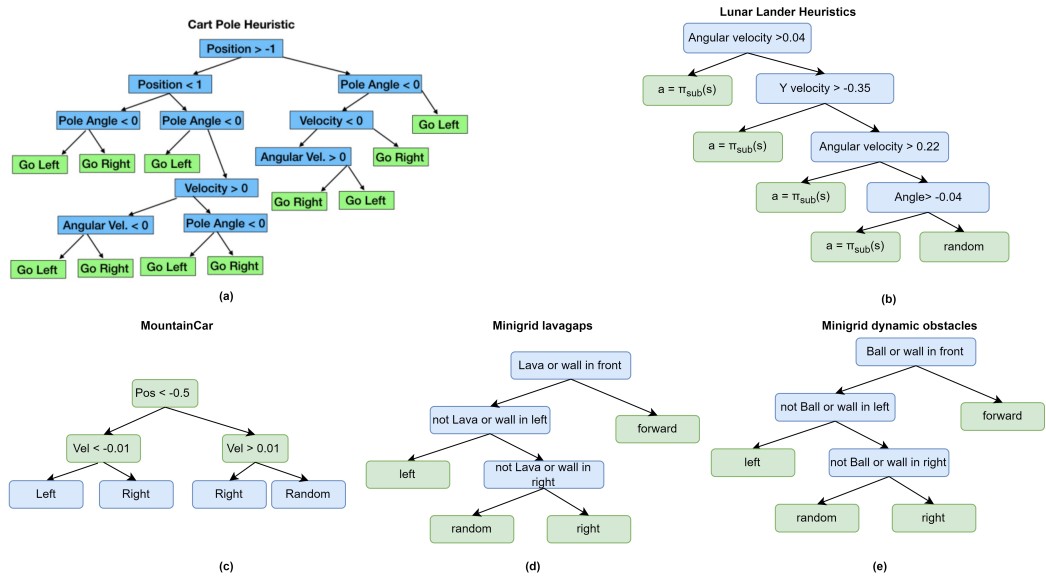

Figure 10: Domain knowledge trees for a) CartPole-v1 b) LunarLander-v2 c) MountainCar-v0 d) MiniGrid-LavaGapS7-v0 e) MiniGrid-Dynamic-Obstacles-Random-6x6-v0 environments

# F  RELATED WORK CONTINUED

Knowledge distillation is a well-embraced technique of incorporating additional information in neural networks and has been applied to various fields like computer vision (Xie et al., 2020; Sohn et al., 2020), natural language processing (Devlin et al., 2018; Tang et al., 2019), and recommendation systems (Tang & Wang, 2018). (Hinton et al., 2015) introduced the concept of distilling knowledge from a complex, pre-trained model (teacher) into a smaller model (student). In recent years, researchers have explored the integration of rule-based regularization techniques within the context of knowledge distillation. Rule regularization introduces additional constraints based on pre-

Table 4: Average reward [↑] obtained during online evaluation over 3 seeds on Minigrid environments

| ENVIRONMENT | $\mathcal{D}$ | BC D | BCQ D | CQL D | **ExID** |
|---|---|---|---|---|---|
| MINIGRID DYNAMIC RANDOM6X6 | 0.50 ± 0.08 | 0.59 ± 0.07 | 0.24 ± 0.22 | 0.14 ± 0.1 | 0.79 ± 0.07 |
| MINIGRID LAVAGAPS 7X7 | 0.27 ± 0.09 | 0.29 ± 0.11 | 0.26 ± 0.1 | 0.28 ± 0.12 | 0.46 ± 0.13 |

defined rules, guiding the learning process of the student model (Hu et al., 2016; Yuan et al., 2020). These techniques have shown to reduce overfitting and enhance generalization (Tang et al., 2019). Knowledge distillation is also prevalent in the field of RL (Zheng et al., 2021) and offline RL (Tseng et al., 2022). Contrary to prevalent teacher-student knowledge distillation techniques, our work does not enforce parameter sharing among the networks. Through experiments, we demonstrate that a simple regularization loss and expected performance-based updates can improve generalization to unobserved states covered by domain knowledge. There are also no constraints on keeping the same network structure for the teacher, paving ways for capturing the domain knowledge into more structured networks such as Differentiable Decision Trees (DDTs).

# G NETWORK ARCHITECTURE AND HYPER-PARAMETERS

We follow the network architecture and hyper-parameters proposed by (Schweighofer et al., 2021) for all our networks, including the baseline networks. The teacher BC network $\pi_\omega^t$ and Critic network $Q_s^\theta(s, a)$ consists of 3 linear layers, each having a hidden size of 256 neurons. The number of input and output neurons depends on the environment's state and action size. All layers except the last are SELU activation functions; the final layer uses linear activation. $\pi_\omega^t$ uses a softmax activation function in the last layer for producing action probabilities. A learning rate of 0.0001 with batch size 32 and $\alpha = 0.1$ is used for all environments. MC dropout probability of 0.5 and number of stochastic passes T=10 have been used for the critic network. The uncertainty check is performed every 15 episodes after the warm start to avoid computational overhead. The hyper-parameters specific to our algorithm for OpenAI gym are reported in Table G. The hyper-parameters specific to our algorithm for Minigrid environments are reported in Table 6. For SalesPromotion and Simglucose tasks we used standard hyperparameters of CORL(Tarasov et al., 2022) library with $\lambda = 0.5$ and $k = 30$.

Table 5: Hyperparameters for openAI gym environments

| HYPERPARAM | MOUNTAINCAR | | | CARTPOLE | | | LUNAR-LANDER | | |
|---|---|---|---|---|---|---|---|---|---|
| DATA TYPE | EXPERT | REPLAY | NOISY | EXPERT | REPLAY | NOISY | EXPERT | REPLAY | NOISY |
| $\lambda$ | 0.5 | 0.5 | 0.5 | 0.5 | 0.5 | 0.5 | 0.5 | 0.5 | 0.5 |
| $k$ | 30 | 30 | 30 | 30 | 30 | 30 | 30 | 30 | 30 |
| $\pi_\omega^t$ LR | $1e^5$ | $1e^5$ | $1e^5$ | $1e^2$ | $1e^2$ | $1e^2$ | $1e^4$ | $1e^4$ | $1e^4$ |
| TRAINING STEPS | 42000 | 36000 | 36000 | 30000 | 17000 | 17000 | 18000 | 18000 | 18000 |

# H EFFECT OF $k$ AND $\lambda$ AND EVALUATION PLOTS

We empirically evaluate the effect of $\lambda$ In Fig 11 and $k$ in Fig 12. We believe these parameters depend on the quality of $\mathcal{D}$. For the given $\mathcal{D}$ in the environments we empirically observe, $\lambda = 0.5$ generally performs well, except for Minigrid environments where $\lambda = 0.1$ works better. Increasing the warm start parameter $k$ generally increases the initial performance of the policy, allowing it to

Table 6: Hyper-parameters for Mini-grid environments for replay dataset

| Environment | DynamicObstRandom6x6-v0 | LavaGapS7v0 |
| --- | --- | --- |
| $\lambda$ | 0.1 | 0.1 |
| $k$ | 30 | 30 |
| $\pi_\omega^t$ lr | $1e^4$ | $1e^4$ |
| training steps | 5000 | 10000 |

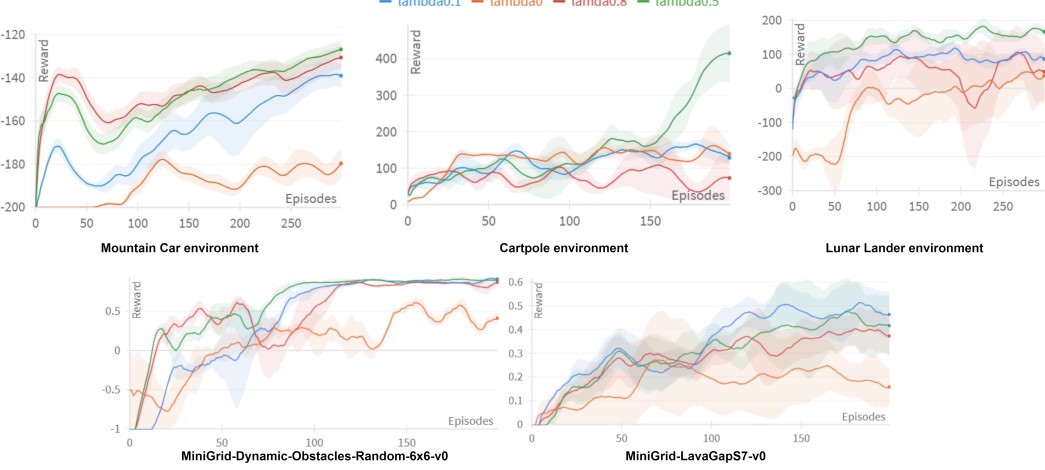

Figure 11: Effect of $\lambda$ on the performance of ExID for different environments expert datasets.

learn from the teacher. Meanwhile, no warm start adversely affects policy performance as the critic may erroneously update the teacher. From empirical evaluation, we observe that $k = 30$ gives a reasonable start to the policy. All the evaluation plots are shown in Fig 13, where it can be observed that ExID performs better than baseline CQL.

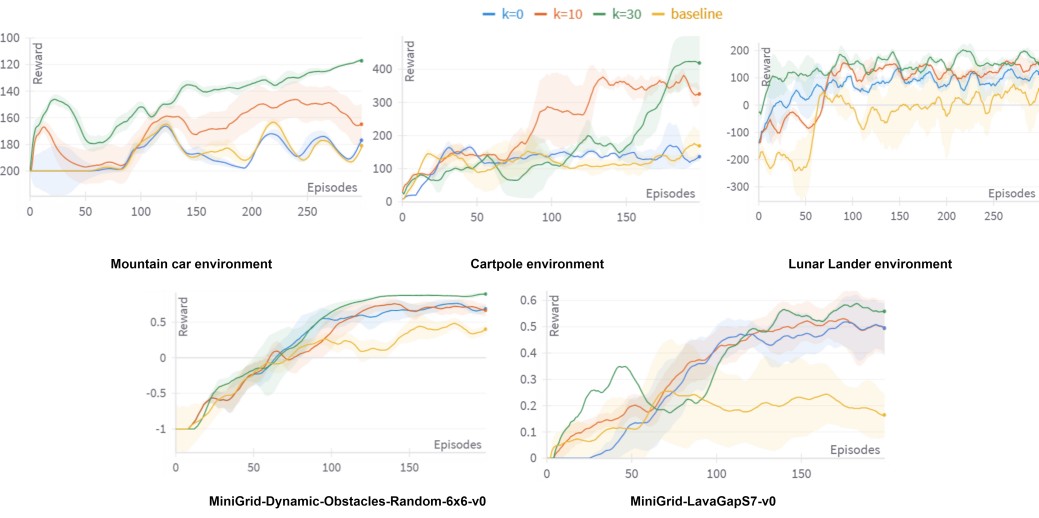

Figure 12: Effect of $k$ on the performance of ExID for different environments expert datasets.

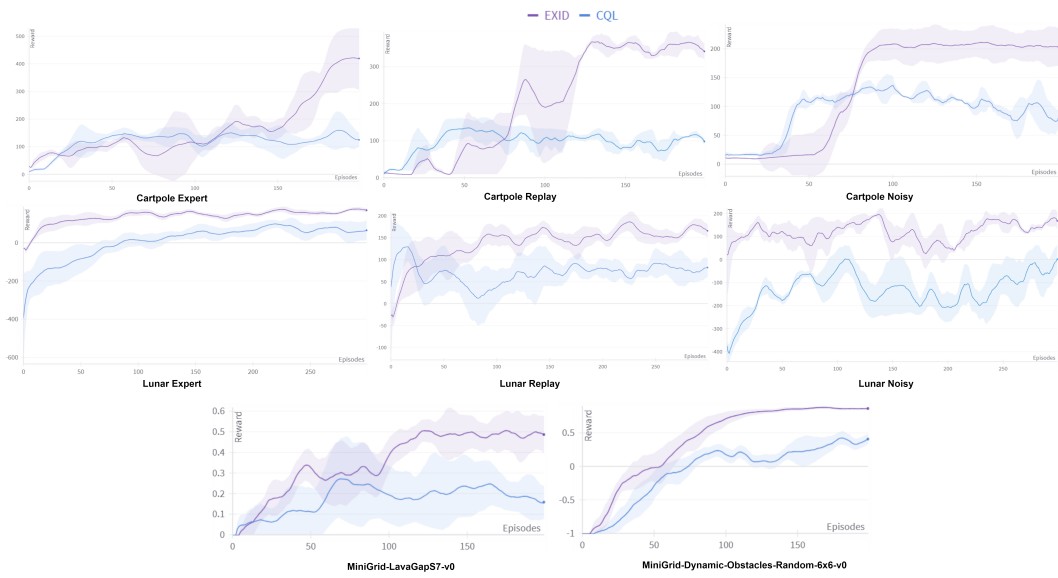

Figure 13: Evaluation plots of CQL and EXID algorithms for Cartpole, Lunar-Lander, and Minigrid environments using different data types and seeds reported in the main paper Table 5.1.

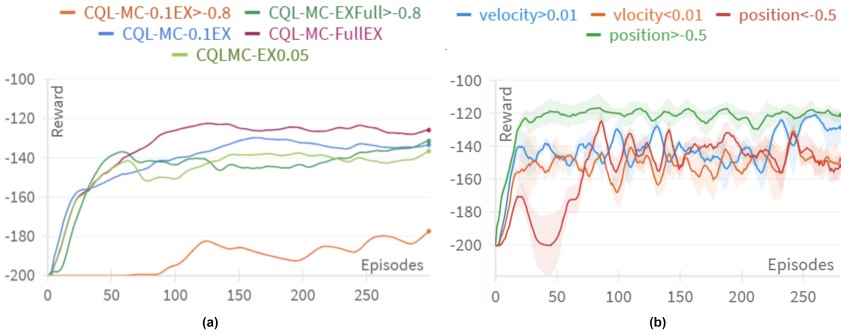

Figure 14: (a) The effect of data reduction and removal on baseline CQL visualized on Mountain Car Environment (b) Performance of ExID on removing different parts of the data based on nodes of Fig 10 (c) from Mountain Car expert dataset

# I DATA REDUCTION DESIGN AND DATA DISTRIBUTION VISUALIZATION OF REDUCED DATASET

In this section, we discuss the intuition behind our data-limiting choices. We also visually represent selected reduced datasets for the OpenAI gym environments.

**Reducing transitions from the dataset:** For all datasets, 10% of the data samples were extracted from the full dataset. This experimental design choice is based on the observation shown in Fig 14 (a). Performance degrades on reducing samples to 0.1% of the dataset and reduces further on reducing samples to 0.05% of the dataset. However, this drop is not substantial. The performance also reduces on removing part of the dataset from the full dataset with states $> -0.8$. However, the worst performance is observed when both samples are reduced and data is omitted, attributing to accumulated errors from probability ratio shift contributing to an increase in generalization error. Our methodology aims to address this gap in performance.

**Removing part of the state space:** Due to the simplicity of the Mountain-Car environment, we analyze the Mountain-Car expert dataset to show the effect of removing data matching state conditions of the different nodes in the decision tree in Fig 10 (c). The performance for each condition is summarised in Table 7. The most informative node in the tree is position $> -0.5$; removing states matching this condition causes a performance drop in the algorithm as the domain knowledge regularization does not contribute significant information to the policy. Similarly, removing data with velocity $< 0.01$ causes a performance drop. However, both performances are higher than the baseline CQL trained on reduced data. Based on this observation, we choose state removal conditions that preserve states matching part of the information in the tree such that the regularization term contributes substantially to the policy. Fig 15 shows the data distribution plot of 10% samples extracted from mountain car replay and noisy data with states $> -0.8$ removed. Fig 16 shows visualizations for 10% samples extracted from expert data with velocity $> -1.5$ removed. Fig 17 shows visualizations for 10% samples extracted from expert data with lander angle $< -0.04$ removed.

Table 7: Performance of ExID on removing different parts of the data based on nodes of Fig 10 (c) from Mountain Car expert dataset

| Position>-0.5 | Position<-0.5 | Velocity>0.01 | Velocity<0.01 |
|---|---|---|---|
| -121.89 ± 7.69 | -151 ± 13.6 | -128.48 ± 11.84 | -147.80 ± 5.01 |

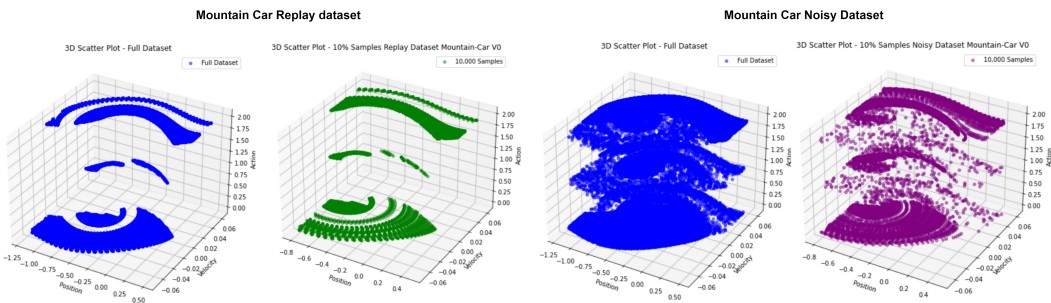

Figure 15: Data distribution of reduced dataset compared to the full dataset for mountain replay and noisy data

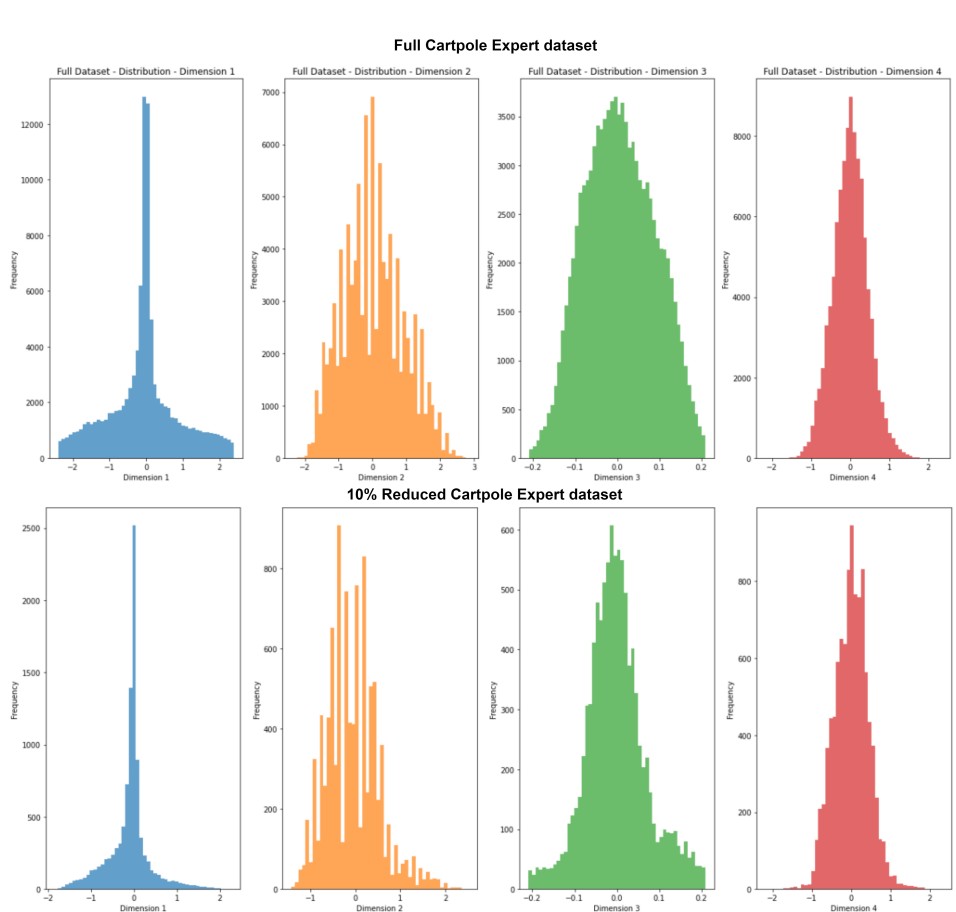

Figure 16: Data distribution of reduced cart pole expert dataset compared to the full dataset

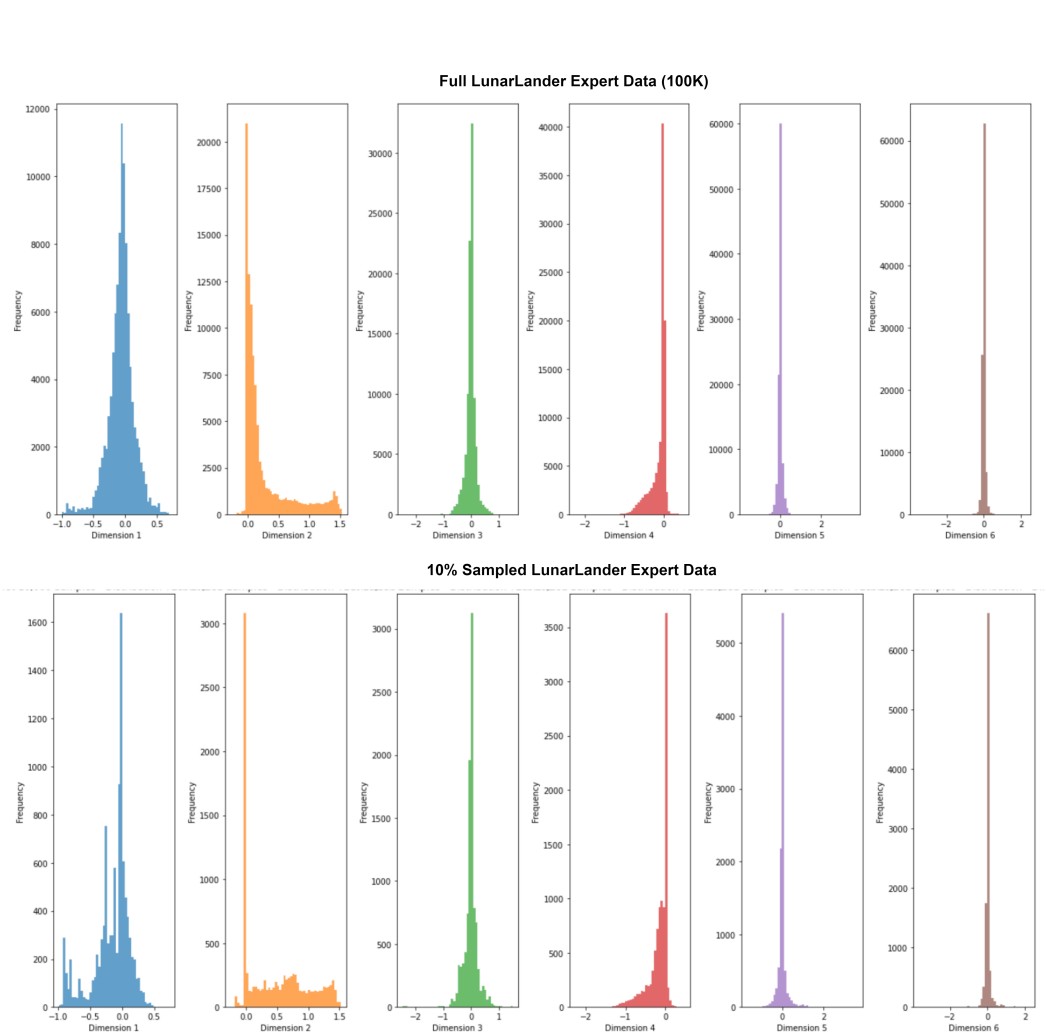

Figure 17: Data distribution of reduced LunarLander expert dataset compared to the full dataset

