# OpenReview forum: "ExID: Offline RL with Intuitive Expert Insights in Limited-Data Settings"
_ICLR.cc/2025/Conference — ICLR 2025 Conference Withdrawn Submission_

### Official Review · Reviewer_frm3 · 2024-10-25

**Soundness:** 1
**Presentation:** 2
**Contribution:** 2
**Rating:** 3
**Confidence:** 4

**Summary:**

This paper proposes an offline RL algorithm, ExID, that can leverage domain knowledge-based heuristic rules to enhance small-sample performance. The proposed algorithm can be seen as combining conservative Q-learning (CQL) and Uncertainty Weighted Actor-Critic (UWAC), and adds a CQL-style domain knowledge-based regularize, i.e., push up Q-values for actions that comply with the domain knowledge and push down Q-values for policy-generated actions. For detailed, comments please refer to strengths and weaknesses.

**Strengths:**

- Leveraging domain knowledge to enhance sample efficiency in offline RL is meaningful.
- The idea of modeling domain knowledge as tree-based heuristic rules and incorporating them using CQL-style value regularization is interesting.
- The paper is easy to read. The results show good performance for tasks that have domain knowledge-based rules.

**Weaknesses:**

- The literature review is insufficient, missing many recent offline RL works that focus on sample efficiency and OOD generalization.
- The proposed method has very stringent requirements for the domain knowledge input. Although we can design domain knowledge trees for simple tasks, it is not practical for complex tasks. Typically, in many real-world tasks, we might only be able to write very few heuristic rules for a task, which can only sparsely cover the state-action space. This severely limits the applicability of the proposed method for a wide range of problems. In the experiments, the authors are also only able to evaluate their method in very simple task environments.
- When training the teacher policy $\pi_t^w$, a random action is used if a state is not covered by the domain knowledge rules. This can be very problematic, as it will inevitably lead to sub-optimality when regularizing the learned policy. Moreover, if only a small set of domain knowledge rules is given (common in most practical problems), then it will cause the teacher policy to learn lots of random behavior, which could damage policy learning.
- The algorithm is essentially a combination of CQL and UWAC, with an additional value regularization that penalizes the value for actions that are not consistent with the domain knowledge. Since CQL is already very conservative and does not generalize well. Adding additional conservative regularization will further distort the value function, which could impact the optimality of the learned value.
- Propositions 4.2 and 4.3 are meaningless and have huge theory-practice gaps. The proposed method combines both CQL and the uncertainty-based method, which makes it almost impossible to draw any reliable theoretical conclusion on the learned value function and the policy. In the proof of Proposition 4.2, the underlying analytical tools used work for any policy, the only thing related to the proposed method is the **assumptions** $\rho_{\hat{\pi}}=\rho_{\pi_{u}}-\Delta_i$ and $\rho_{\hat{\pi}}=\rho_{\pi_{u}}-\Delta_o$, with $\Delta_i$ and $\Delta_o$ assumed to be positive, as well as $Q^*(s,\hat{\pi}(s))-Q^*(s,\pi_u(s)) \approx 0$ for in distribution actions. First, these assumptions have no guarantees from your algorithm. Second, to prove something, you simply cannot introduce some arbitrary assumptions to achieve your purpose. Proposition 4.3 is also trivial and not very meaningful. It only relates to the optimal value function $Q^*$, rather than the learned value function $Q^{\theta}$. You can only learn $Q^{\theta}$, and the proposition provides no information to tell if the teacher policy indeed helps to improve the learned $Q^{\theta}$.
- There are also some inconsistencies in the symbols used in the paper. For example, the teacher policy is expressed as $\pi_t^w$ in the text, but written as $\pi_w$ in Figure 2. The action value function is written as $Q_s^{\theta}$ in the text, but written as $Q_{\theta}$ in Figure 2.

**Questions:**

- How can the method be used if only limited domain knowledge rules are provided?
- Is it possible to not use random action for cases that are uncovered by domain knowledge rules?

---

> ### Author Response · Authors · 2024-11-15
> **Response to Reviewer frm3**
>
> W1:  Literature review is insufficient
>
> - We have made efforts to cover the most well-known methods in offline RL, as detailed in Section 2 of the main text. Additionally, we have provided comparisons with recent offline RL works, such as SCQ, A2PR, and CPED, in Sections D and F of the Appendix.
> -  If the reviewer could kindly point out any relevant references that we may have missed, we would be happy to include them in the revised manuscript.
>
> W2: Stringent requirements for the domain knowledge input
>
> - As the reviewer correctly points out, "in many real-world tasks, we might only be able to write very few heuristic rules for a task, which can only sparsely cover the state-action space." This is precisely the problem setup addressed in this manuscript.
> - For example in the sales promotion task in line 398 the domain knowledge only covers states with order_number > 60 and avg_fee > 0.8. Also the rules considered in experiments are imperfect and updated during the offline policy training which extends applicability to many domains where only partial heuristic rules are available.
> - Beyond the six standard discrete Gym environment tasks, the mythology is also validated on human generated data from business domain  (**Sales Promotion**) and healthcare task (**Simglucose**). These tasks are not simple and exhibit intricacies of real world problems. Please refer to [1] and [2] for the description of these tasks.
>
> [1] **Simglucose**: Jinyu Xie. Simglucose v0.2.1 (2018) [Online]. Available: [https://github.com/jxx123/simglucose](https://github.com/jxx123/simglucose).
>
> [2] Qin, Rong-Jun, et al. "NeoRL: A near real-world benchmark for offline reinforcement learning." _Advances in Neural Information Processing Systems_ 35 (2022): 24753-24765.
>
> W3: Problem induced by random actions for states not covered by domain knowledge
>
> This is not problematic as Q values are only regularized when the states match the domain knowledge as depicted in Eq 3 $\mathcal{L}_r(\theta) = \underbrace{\mathbb{E}_{s \sim \mathcal{B}_r \land s \models \mathcal{D}}}_{\text{states matching domain rule}} \underbrace{[Q_s^\theta(s, a_s)-Q_s^\theta(s,a_t)]^2}_{\text{Q regularizer}}$.Random actions are merely placeholders in the behavior cloning (BC) policy during training from domain rules and do not affect the Q-value regularization process.
>
> W4: The algorithm is essentially a combination of CQL and UWAC
>
> - We respectfully disagree with this point. In practice, it is rarely possible to obtain perfect domain knowledge. Therefore, our algorithm updates the initial teacher network ($\pi_t^\omega$) domain knowledge using Eq 5, 6 and 7 during training.  This mechanism of knowledge distillation has not been studied before for offline RL.
> - The proposed algorithm deviates from standard CQL or UWAC, which primarily promote conservatism for out-of-distribution (OOD) actions in states observed within the dataset. The key contribution of **ExID** lies in distilling knowledge for OOD actions in states that rarely or never appear in the dataset, addressing a critical gap that existing methods do not tackle.
>
> W5: Propositions 4.2 and 4.3
>
> - We acknowledge that combining CQL with uncertainty-based methods complicates theoretical guarantees, and we agree that certain assumptions in Propositions 4.2 lacks direct support from the algorithm. These assumptions were intended as idealizations to simplify the analysis and provide preliminary insights into how reasonable domain knowledge regularization can lead to performance improvements. They are not intended to be the major contribution of the paper.
> - In response to your concerns, we have moved proposition 4.2 completely to the appendix in the revised manuscript and have explicitly labeled the assumptions as simplifying in line 273. We have also removed proposition 4.3. We hope this adjustment provides a more transparent and grounded presentation of our results.
>
> W6: There are also some inconsistencies in the symbols used in the paper
>
> Thank you for pointing these out. We have corrected in revision.
>
> Q1:  How can the method be used if only limited domain knowledge rules are provided?
>
> Please refer to our response for W2.
>
> Q2:   Is it possible to not use random action for cases that are uncovered by domain knowledge rules?
>
> Please refer to our response for W3.

---

> > ### Comment · Reviewer_frm3 · 2024-11-24
> > **Thanks for the response**
> >
> > I thank the reviewer for the response. However, most of my previous concerns remain unsolved, hence I choose to keep my score unchanged. Specifically,
> >
> > - Regarding the response to W1, the newly added offline RL baselines are not really focused on sample efficiency and OOD generalization. The authors need to at least compare with the SOTA small-sample offline RL algorithm TSRL [1] or recent works that explored generalization issues in offline RL, such as [2].
> >
> > [1] Look Beneath the Surface: Exploiting Fundamental Symmetry for Sample-Efficient Offline RL. NeurIPS 2023.
> >
> > [2] When Data Geometry Meets Deep Function: Generalizing Offline Reinforcement Learning. ICLR 2023.
> >
> > - Regarding the response to W2, note that all your experiments are conducted on simple tasks (e.g., mountain car, cartpole, lunar lander), with extremely low action dimensions. Even the sales promotion task referred to by the author has only two actions. Of course for these low action dimension tasks, you can write heuristic rules, but for most real-world or more complex tasks with higher action dimensions, especially those having complex inter-connected relationships among actions, it is simply unrealistic to construct the required domain knowledge input needed in this method.
> >
> > - Regarding the response to W3, I still have concerns regarding introducing random actions for the teacher policy.
> >
> > - Regarding the response to W4, I agree that there are some differences compared with existing methods, but I still think the modification and additional insights on the policy learning part are somewhat incremental.

---

### Official Review · Reviewer_N2mw · 2024-10-29

**Soundness:** 2
**Presentation:** 1
**Contribution:** 2
**Rating:** 3
**Confidence:** 3

**Summary:**

This paper work on incorporating domain knowledge into offline RL to improve OOD performance.
The method appears to be based on DQN, but incorporates a "teacher" policy network which is used to regularize the argmax Q-function policy.

**Strengths:**

The use of domain knowledge in RL is worthy of study.

The results seem to indicate some empirical advantage of the method over baselines (although not consistently so).

**Weaknesses:**

**Unconvincing motivation and contributions**
This seems to be an avenue of work that has received very limited attention, and I would appreciate a more thorough motivation for both the problem setting and the approach proposed.

The environments studied seem pretty simple, and I'm not convinced by the baselines.
I'm skeptical that this approach can scale, as it is based on strong domain knowledge.
The practical utility on simple environments is also unclear -- can the authors make a more clear and compelling case of when and why they expect this approach to yield practical value?
I would also expect more discussion of how well this approach can scale to more complicated environments, and what sort of future work might help answer that question.

**Other presentation issues**
- I found the presentation of the algorithm confusing and presentation to be weak overall.
- should use citep
- Definition 4.1 is confusing.
-- I think the notation in the first condition (bullet point) is used incorrectly.
-- The 2nd bullet point seems redundant (being guaranteed to hold if the first condition holds).


**Inadequate treatment of model-based RL**
The work needs a more thorough comparison with model-based offline RL methods.
Currently, there is only one experiment reported (in insufficient detail) in the appendix.
I would expect these baselines to be run for all of the environments and reported in main text.
As such, I'm not convinced the authors have made a fair effort to tune and compare their methods with baselines.
Furthermore, the point of comparison is MOPO, a method from 2020 -- are there more modern methods which ought to be considered?

The submission also states that for such methods "performance highly depends on the accuracy of the learned dynamics."  But this claim is not supported.  These methods are meant to keep policies from seeking out states where the learning dynamics are inaccurate, and thus, we might expect performance to degrade gracefully in the presence of inaccurate dynamics.  On what basis are the authors claiming otherwise?

**Questions:**

Are there more modern offline RL methods which ought to be considered?

Can your approach be viewed as a form of actor-critic?  Would an actor-critic method be more performant?

What are some alternative methods for

Does D specify a complete policy (I.e. does it have support on the entire state space)?  If so, I think the notation should be pi_D.

---

> ### Author Response · Authors · 2024-11-15
> **Response to reviewer N2mw**
>
> Thank you for your valuable feedback. Please find our response below:
>
> W1: Unconvincing motivation and contributions:
>
> - The motivation for this work is rooted in real-world use cases where heuristic-based domain knowledge is readily available but offline data is limited. This scenario applies to many practical problems, such as resource allocation in business, healthcare, and autonomous driving, where domain-specific heuristics are common. The limited attention given to this problem setup provides a strong motivation for research in this direction.
>
> - To demonstrate the practical effectiveness of our method, we validated it not only on six standard discrete Gym environment tasks but also human generated data from business domain  (**Sales Promotion**) and healthcare task (**Simglucose**). These tasks are not simple and exhibit intricacies of real world problems. Please refer to [1] and [2] for the description of these tasks.
>
> - We have shown comparison with existing model free RL baselines that use domain knowledge  like CQL with safey expert (CQL-SE) and ensemble of offline RL methods and domain knowledge.
> - For continuous domain tasks we have also compared with recent offline RL baselines SCQ,  A2PR, CPED (Section D). We will be happy to compare with any other baseline that the reviewer might suggest.
>
> [1] **Simglucose**: Jinyu Xie. Simglucose v0.2.1 (2018) [Online]. Available: [https://github.com/jxx123/simglucose](https://github.com/jxx123/simglucose).
>
> [2] Qin, Rong-Jun, et al. “NeoRL: A near real-world benchmark for offline reinforcement learning.” _Advances in Neural Information Processing Systems_ 35 (2022): 24753-24765.
>
> W2 : Other presentation issues
>
> - We have restructured parts of the manuscript to make the presentation better. Please find the summary in the global comment. We are happy to revise for any additional suggestion the reviewer might have.
> - Thank you for suggesting citep we have used this for the updated manuscript.
> - In Def 4.1 The first condition discusses about missing states. This means that there are states $s$ in $B$ for which no transitions $(s,a,s')$ exists in $B_r$.
> - The second conditions denotes reduced sampled for some transition. This means that while the transition $(s,a,s')$ exists in both $B$  and $B_r$, there are fewer instances of it in $B_r$,compared to $B$. This is not guaranteed to hold given the first condition. For example let B contain 10 sample of t1: (s_1,a_1,s_2) and 5 samples of t2: (s_1,a_2,s_3) just removing s_2 from $B_r$ does not denote reducing samples for t2.
>
> W3 : Inadequate treatment of model-based RL
>
> - The proposed method is specifically designed to address the generalization gap of offline model-free RL methods in limited data settings. In this context, a direct comparison with different model-based RL methods may not be entirely fair to the proposed approach.
> -  Both model-free and model-based RL have their respective caveats. As discussed in the original MOPO paper, dynamics inaccuracies in poorly covered regions of the state space can still lead to performance degradation, even when uncertainty penalties are applied. The paper highlights that substantial model inaccuracies, particularly in out-of-distribution (OOD) regions, can result in performance losses due to compounding errors in hypothetical rollouts. Given the limited data setting we address, learning an accurate dynamics model is inherently challenging. The paper does not make claims beyond this limitation. The validity of this claim is empirically supported by results on the human-generated **Sales Promotion (SP)** dataset.
> - Regarding tuning of baselines, all baselines have been taken from standard implementations associated with the respective papers. We cited the sources in the manuscript, and further details can be found in our code, which is provided in the supplementary material.
>
> Q1. Are there more modern offline RL methods which ought to be considered?
>
> We have compared the discrete tasks with existing discrete offline RL algorithms and the continuous tasks with recent algorithms, such as SCQ, A2PR, CPED. If the reviewer has suggestions for additional relevant baselines, we would be happy to include a comparative analysis in future revisions.
>
> Q2. Can your approach be viewed as a form of actor-critic? Would an actor-critic method be more performant?
>
> - Only inclusion of an actor critic network does not improve performance in limited data setting.  The continuous domain tasks in our study utilize actor-critic networks, and Figure 3c illustrates the performance gap.

---

> > ### Author Response · Authors · 2024-11-15
> > **Response continued**
> >
> > - The primary reason for this gap is that current offline RL methods typically only apply out-of-distribution (OOD) action correction for states present in the dataset. They lack mechanisms for correcting actions in OOD states outside the dataset, which leads to performance degradation when data is limited. In contrast, **ExID** distills domain knowledge to correct OOD actions for such states, thereby achieving performance improvements over baseline methods.
> > - Our method is applied to the continuous domain which uses both actor and critic network by using the regularization in Eq 4 : $\mathcal{L}(\theta) = \mathcal{L}cql(\theta)  + \lambda
> > E_{s \sim \mathcal{B}r \land s \models \mathcal{D}} [Q_s^\theta(s, a_s)-Q_s^\theta(s,a_t)]^2$ during critic $(Q_s^\theta)$ training for continuous domain and using actions from actor network $(\pi_s)$ for cross entropy loss in Eq 7 : $\mathcal{L}(\omega) = -\sum_{s \models D} (\pi_t^\omega(s)log(\pi_s(s)))$. We also show this via experiments in Sec 5.3.
> >
> > Q3. What are some alternative methods for
> >
> > We believe this question is incomplete. We would be happy to provide a detailed response if the reviewer could clarify or provide the complete question.
> >
> > Q4. Does D specify a complete policy (I.e. does it have support on the entire state space)? If so, I think the notation should be pi_D.
> >
> > D specifies domain knowledge which is in the form of S-> A mapping. These are heuristic rules that cover a subset of the state space but do not necessarily define a policy for all states. Hence we denote the domain rules as D.

---

> > ### Comment · Reviewer_N2mw · 2024-11-18
> > **Definition 4.1**
> >
> > * Please address my point about notation in the first bullet point.
> > * I maintain that the 2nd bullet point is redundant: if there exists a state which is present in B and not present in B_r, then the number of transitions from that state in B_r is 0, satisfying the criteria of the 2nd bullet point.

---

> > > ### Author Response · Authors · 2024-11-18
> > > **Response to reviewer N2mw**
> > >
> > > Thank you for taking the time to read our rebuttal and for your valuable feedback. Below, we provide responses to your concerns:
> > >
> > > Regarding Def 4.1
> > > - Could the reviewer kindly specify which part of the notation is used incorrectly? This would allow us to clarify further and make the necessary revisions.
> > > - While we understand the reviewer's perspective, we respectfully reiterate that the second bullet point is not redundant. As outlined in our rebuttal with an example, Point Two addresses the removal of specific transitions rather than states. To clarify:
> > > - Suppose \( B \) contains 10 samples of \( t_1: (s_1, a_1, s_2) \) and 5 samples of \( t_2: (s_1, a_2, s_3) \).
> > > -- Point One states that certain states, such as \( s_2 \), are removed in \( B_r \). This removal naturally eliminates all transitions leading to \( s_2 \) in \( B_r \).
> > > -- Point Two, however, states that for states present in both \( B \) and \( B_r \), some transitions are selectively removed. For example, in \( B_r \), there may be only 3 instances of \( (s_1, a_2, s_3) \) instead of 5.
> > > This demonstrates that Point Two captures a distinct concept that cannot be reduced to Point One.
> > >
> > > Regarding Model based RL
> > >
> > > - We appreciate your perspective and understand the importance of fair comparisons. However, we maintain that comparing a model-free RL method with all model-based RL methods may not provide meaningful insights given the fundamental differences in assumptions and design. Instead:
> > > - If the reviewer has a specific model-based RL baseline that they believe would strengthen the paper, we are open to conducting additional experiments.
> > > - We also reiterate that we have used the official implementation from the benchmark paper, which has been tuned for the environment under consideration. While the uncertainty penalty may need tuning for a new environment, our choice ensures a fair and reproducible comparison in the current setting.

---

> > > > ### Comment · Reviewer_N2mw · 2024-11-18
> > > > **Response**
> > > >
> > > > Notation: It's hard to tell what you're trying to say, because the notation just doesn't parse for me.
> > > >  I think the "and" symbol is trying to say "such that"?  I'm really not sure.  The statement still doesn't parse properly with that replacement.  isn't B only containing triples?  so "s' in B" is already a nonsense clause.
> > > >
> > > > I'm not going to engage further on the logical implications of the conditions, but one more try: 0 < N for all positive numbers N, so the idea that there are some transitions that are less represented in B_r than in B is covered by the condition that there are some transitions that are not represented in B_r at all (but are in B).
> > > >
> > > > RE experiments: At a minimum, I am expecting the model-based method MOPO to be evaluated across all environments.  Tuning the penalty is obviously necessary for the reasons you described, and is something which is evaluated in the MOPO paper.

---

> > > > > ### Author Response · Authors · 2024-11-19
> > > > > **Response to reviewer N2mw**
> > > > >
> > > > > We thank the reviewer for the insightful comments. We have carefully considered the feedback and made revisions to address the concerns raised. Below, we outline our changes and clarifications:
> > > > >
> > > > > Notation:
> > > > >
> > > > > We understood the logical inconsistency in Def 4.1 and have rewritten it as follows in the revision:
> > > > >
> > > > > Let $\mathcal{B}$ be the original offline reinforcement learning buffer, represented as a multiset of transitions $(s, a, s')$. Each transition $(s, a, s')$ appears a certain number of times in $\mathcal{B}$, which we denote as $N_{\mathcal{B}}(s, a, s')$.
> > > > >
> > > > > The reduced buffer $\mathcal{B}_r$ is a sub-multiset of $\mathcal{B}$, such that the number of occurrences of any transition $(s, a, s') $ in $\mathcal{B}_r$, denoted $N_Br(s, a, s')$, satisfies: $N_Br(s, a, s') \leq N_B(s, a, s')$
> > > > >
> > > > > Comparison with MOPO:
> > > > >
> > > > > Unfortunately, no standard implementation of MOPO exists for discrete datasets, which prevented us from applying it directly to our discrete environment experiments. However, we have implemented MOPO for the two continuous domains in our study, using the baseline implementation from (https://github.com/polixir/NeoRL/blob/benchmark/benchmark/OfflineRL/offlinerl/algo/modelbase/mopo.py) which has been tuned for the tasks discussed in the paper. The corresponding results have been reported in the revised manuscript.

---

> > ### Comment · Reviewer_N2mw · 2024-11-18
> > **RE model-based methods**
> >
> > I maintain that the comparisons here are not nearly thorough enough.
> >
> > The uncertainty penalty certainly ought to be tuned for a new environment, and I believe model-based methods deserve equal treatment with the other baselines, not just one experiment in the appendix.

---

### Official Review · Reviewer_341n · 2024-11-02

**Soundness:** 2
**Presentation:** 2
**Contribution:** 3
**Rating:** 6
**Confidence:** 3

**Summary:**

In this work, the authors propose a new transfer offline RL method ExID, which transfer the learned policy of the source domain into the target domain with a expert system and limit data of target domain.

**Strengths:**

- This work forces on a new and important setting. This setting is make sence because the less the data is, the more the risk is.
- This work proposes a simple but effect architecture, which update the target network and the teacher at the same time. This architecture let the method can be used with expert systems with different qualities.
- The final result is good especially when the target data is noisy.

**Weaknesses:**

- The presentation of this work is limit. For example, in Section 3, there is "(s,a,s')\in B, then s'\in B". What exactly the structure of the buffer is? Also, the Eq. 2 is hard to understand at the first time.
- The environments used in this work are all simple and visible. Can we propose a useful expert system for more complex environments, such as halfcheetah?

**Questions:**

- This work leverages a uncertainty based method to update the teacher network. Is this a stable standard? How many times the teacher network is updated in the learning process?
- How many feasible s of  Proposition 4.3?

---

> ### Author Response · Authors · 2024-11-15
> **Response to Reviewer 341n**
>
> Thank you for your valuable feedback. Please find our response below:
>
> W1:   The presentation and Eq. 2.
>
> - The buffer B contains tuples (s, a, s', r) representing state, action, next state, and reward, as is standard in offline RL buffers. The first point states that some states s' the exist in B are not present in reduced buffer $B_r$. We have revised the explanation of of Eq 2 in the manuscript providing clarity for Def 4.1.
>
> W2: Use of simple environments
>
> - We would like to emphasize the proposal is validate on two real world task  sales promotion (business domain) and Simglucose (healthcare) tasks. These tasks are not simple and exhibit intricacies of real world problems. Please refer to [1] and [2] for the description of these tasks.
>
> - While we acknowledge that our methodology relies on domain knowledge—which may be challenging to design for high-dimensional robotics tasks such as those in d4rl (e.g., HalfCheetah)—domain knowledge is often more accessible in business and healthcare contexts. We believe that this type of problem setup deserves research attention, as offline RL is not solely intended for robotics. Its application to real-world business domains is a promising and motivating area of study.
> - If the reviewer has any suggestions of datasets relevant to the proposed problem domain we shall be happy to conduct experiments.
>
> [1] **Simglucose**: Jinyu Xie. Simglucose v0.2.1 (2018) [Online]. Available: [https://github.com/jxx123/simglucose](https://github.com/jxx123/simglucose).
>
> [2] Qin, Rong-Jun, et al. “NeoRL: A near real-world benchmark for offline reinforcement learning.” _Advances in Neural Information Processing Systems_ 35 (2022): 24753-24765.
>
> Q.    How many feasible s of Proposition 4.3?
>
> We apologies for not understanding this question. If the reviewer could clarify we would be happy to answer.

---

> > ### Comment · Reviewer_341n · 2024-11-15
> > **The updating time of the teacher network**
> >
> > I think the updating of the teacher network is important for the total learning process. In the article, the teacher network is updated according to the uncertainty, which means that the time of updating is not a stable number. I'd like to know that how many times the teacher network will be updated in each dataset?

---

> ### Author Response · Authors · 2024-11-16
> **Response**
>
> Thank you for clarifying the question. As you mentioned the teacher network updates are driven by uncertainty, meaning the frequency of updates is not fixed and depends on the dataset. We tracked the number of teacher network updates for each environment using one  seed and following the same training protocol. The results are summarized below:
>
> | Environment | count|
> |-------------|------------------ |
> | SalesPromotion         | 7/150|
> | SimGlucose         | 15/200 |
> | MountainCar (Expert)         | 6 /300|
> | MountainCar (Replay)         | 2 /300|
> | MountainCar (Noisy)         | 1 /300|
> | CartPole   (Expert)       | 8/200 |
> | CartPole   (Replay)       | 2/200 |
> | CartPole   (Noisy)       | 2 /200|
> | LunarLander  (Expert)        | 5/300 |
> | LunarLander  (Replay)        | 3/300|
> | LunarLander  (Noisy)        | 1/300|

---

> > ### Comment · Reviewer_341n · 2024-11-18
> >
> > Thank you for your clear and quickly answer. However, I wonder that what the count exactly mean? Is it per episode or per batch?

---

> > > ### Author Response · Authors · 2024-11-18
> > > **Response to Reviewer 341n**
> > >
> > > Thank you for your question and apologies for not clarifying the count. The count is recorded after the entire training process. Which means for Sales Promotion task the teacher was updated 7 times in the entire training duration containing 150 episodes. We updated the original table to show the number of episodes per dataset.

---

> > > > ### Comment · Reviewer_341n · 2024-11-18
> > > >
> > > > Well, is this result show that the updating of the teacher network is not so important? How about changing the threshold to increase the updating frequence?

---

> > > > > ### Author Response · Authors · 2024-11-18
> > > > > **Response to Reviewer 341n**
> > > > >
> > > > > Thank you for your question. While the results may suggest that the teacher network is updated relatively infrequently, its role remains crucial in guiding the critic during the early stages of training. As seen in Fig. 4c, a suboptimal teacher can lead to performance degradation, highlighting the significance of updating teacher network.
> > > > >
> > > > > Regarding the frequency for updating, it is true that lowering the warm start parameter would increase the frequency of updates. However, this involves a trade-off: as it might lead to updates based on less reliable critic actions, potentially reducing overall stability as shown in Fig 12 (Appendix). In our setup, the update frequency is automatically governed by Eq. 6, which ensures updates only when the critic's action is both better (higher Q-value) and more certain than the teacher’s. This adaptive mechanism balances the benefits of leveraging new critic insights while maintaining robustness.
> > > > >
> > > > > If the dataset quality improves (e.g., using expert data), the critic is likely to identify better actions more frequently, leading to a higher number of teacher updates, as observed in the expert datasets.

---

### Official Review · Reviewer_msTz · 2024-11-03

**Soundness:** 3
**Presentation:** 2
**Contribution:** 3
**Rating:** 5
**Confidence:** 3

**Summary:**

This paper presents ExID, an offline RL method that enhances policy learning in limited-data settings by using a domain-knowledge-based regularization technique, significantly improving performance over traditional offline RL approaches across diverse datasets.

**Strengths:**

1. Innovative to incorporate the domain knowledge.
2. real-world application in sales promotion dataset and simglucose dataset.

**Weaknesses:**

1. Training the teacher network highly requires domain knowledge.
2. The method is heuristic and plug-and-play in different algorithms.
3. The simulation tasks are simple.

**Questions:**

1. Can this method be applied to a continuous environment? If yes, should the authors try difficult simulated tasks, such as d4rl?
2. Can this method be applied to different base algorithms, e.g., BEAR, IQL.

---

> ### Author Response · Authors · 2024-11-15
> **Response to reviewer msTz**
>
> Thank you for your valuable feedback. Please find our response below:
>
> W1.  Training the teacher network highly requires domain knowledge.
>
> - Yes, we have explicitly acknowledged the dependency on domain knowledge in our problem setting and discussed it as a limitation in the manuscript. However, we would like to emphasize that the domain knowledge used is not perfect; it is heuristic-based and derived from human understanding, which reflects practical constraints in real-world scenarios.
>
>
> W2.  The method is heuristic and plug-and-play in different algorithms.
>
> - We respectfully disagree with this point. The method requires careful regularization of the critic network and an uncertainty-based update of the initial teacher network. We have detailed the effects of each hyperparameter introduced in our approach in Figures 11 and 12 of the appendix (supplementary material).
> - While the initial knowledge is distilled from heuristics, it is further updated during offline training, which is a critical step. This update process is essential for achieving the performance improvements demonstrated in our experiments and ablation studies. To the best of our knowledge this mechanism of knowledge distillation has not been previously studies.
>
> W3. The simulation tasks are simple
>
> - The datasets used in this paper are recognized and utilized by the research community. Beyond the six standard Gym environment tasks, the mythology is also validated on human generated data from business domain  (**Sales Promotion**) and healthcare task (**Simglucose**).
>
> - These tasks are not simple and exhibit intricacies of real world problems. Please refer to [1] and [2] for the description of these tasks.
>
> [1] **Simglucose**: Jinyu Xie. Simglucose v0.2.1 (2018) [Online]. Available: [https://github.com/jxx123/simglucose](https://github.com/jxx123/simglucose).
>
> [2] Qin, Rong-Jun, et al. "NeoRL: A near real-world benchmark for offline reinforcement learning." _Advances in Neural Information Processing Systems_ 35 (2022): 24753-24765.
>
> Q1 Can this method be applied to a continuous environment? If yes, should the authors try difficult simulated tasks, such as d4rl?
>
> - Yes this method is applicable to continuous domain. Both sales promotion and sim glucose tasks are from continuous domain. However, this method is designed for use cases where reasonable domain knowledge is available as stated in our problem setup and limitation.
> - While we acknowledge that our methodology relies on domain knowledge—which may be challenging to design for high-dimensional robotics tasks such as those in d4rl (e.g., HalfCheetah)—such domain knowledge is more accessible in business and healthcare contexts. We believe that this type of problem setup deserves research attention, as offline RL is not solely intended for robotics. Its application to real-world business domains is a promising and motivating area of study.
> -  If the reviewer has any suggestions of datasets relevant to the proposed problem domain we shall be happy to conduct experiments.
>
> Q 2. Can this method be applied to different base algorithms, e.g., BEAR, IQL.
>
> Yes, the method is applicable to other base algorithms with an actor-critic structure, as it primarily adds a critic regularization term.

---

### Official Review · Reviewer_Dypb · 2024-11-04

**Soundness:** 2
**Presentation:** 2
**Contribution:** 2
**Rating:** 3
**Confidence:** 3

**Summary:**

This paper aims to incorporate the domain knowledge in the sparse states where the offline data is not explored much.
Firstly, the teacher policy $\pi_t^w$ is trained by using the domain knowledge in the heuristic manner. And then, the offline policy is trained using a proposed critic-regularization term to enhance performance.

**Strengths:**

- The novel concept: It is not easy to incorporate some general domain knowledge to train the policy. They try to put somehow symbolic control in the discrete action offline RL.
Method

- The critic regularization term: To use domain knowledge in the offline RL setting, the proposed regularization term is intuitive and proper.

**Weaknesses:**

- The method ‘Training Teacher’ is somehow naive, and Eq. (2) is not clear. (Actually, I don’t understance what it is)

- Limited domain: It is not applicable in the continuous action space.

- The hyperparameter $\lambda$, the coefficient constant for critic regularization. There is no exact way to decide the proper value of $\lambda$.

- Updating the Teacher policy: The explanation for updating the teacher policy, especially after line 226, becomes difficult to follow.

- Writing should be enhanced.

**Questions:**

In the regularization term, are there any gradient stop or .detach() (in torch) in Q-network?
Line 128, The definition of target network parameter $\theta'$ is missing.

---

> ### Author Response · Authors · 2024-11-15
> **Response to reviewer Dypb**
>
> Thank you for your valuable feedback. Please find our response below:
>
> W1: The method ‘Training Teacher’ is somehow naive, and Eq. (2) is not clear.
>
> - Our proposal is not centered on training the teacher network itself but rather on demonstrating how domain knowledge effectively contributes to out-of-distribution (OOD) action regularization through critic regularization and teacher updates. We employ behavior cloning to train the teacher network, which is a well-established practice in knowledge distillation. If the reviewer has any suggestions for a better training paradigm, we would be happy to explore them.
> - For Eq 2 we add the following explanation in the revised manuscript:
> We assume that a set of common-sense rules in the form of domain knowledge, denoted as $\mathcal{D}$, is available. This domain knowledge defines a hierarchical mapping from states to actions $(S \to A)$, structured as decision nodes. Each decision node $T_{\eta_i}$ has constraint $\phi_{\eta_i}$ that determines its branching, a Boolean indicator $\mu_{\eta_i}$ selects the branch $(\swarrow or \searrow)$ to follow based on whether the constraint $\phi_{\eta_i}$ is satisfied.
>
> W2: Limited domain
>
> This method is extended to continuous domain by using the regularization in Eq 4 : $\mathcal{L}(\theta) = \mathcal{L}cql(\theta)  + \lambda
> E_{s \sim \mathcal{B}r \land s \models \mathcal{D}} [Q_s^\theta(s, a_s)-Q_s^\theta(s,a_t)]^2$ during critic $(Q_s^\theta)$ training for continuous domain and using actions from actor network $(\pi_s)$ for cross entropy loss in Eq 7 : $\mathcal{L}(\omega) = -\sum_{s \models D} (\pi_t^\omega(s)log(\pi_s(s)))$. We have added this explanation to the revised manuscript (lines 273–275) and also demonstrate it through experiments. The **Sales Promotion** and **Simglucose** tasks reported in Section 5.3 operate on continuous state-action domains.
>
>
> W3: The hyperparameter
>
> - We agree that hyperparameter tuning in offline settings can be challenging. In practice, this can be addressed using methods such as Bayesian Optimization and by observing the behavior of Q-values, as proposed in [2].
> -  $\lambda$ should be selected with peak Q value observed during training. We discuss the effect of different $\lambda$ in Fig 12 (supplement) and empirically show $\lambda$ = 0.5 works well in most setting. We also show the proposed methodology is better than $\lambda$ = 0 in most environments demonstrating robustness to hyperparameters.
>
> [2] Kumar, A., Singh, A., Tian, S., Finn, C., & Levine, S. (2021). A workflow for offline model-free robotic reinforcement learning. arXiv preprint arXiv:2109.10813.
>
> W4 : Updating teacher
>
> We restructured the updating teacher section in the revised manuscript providing more clarity as follows:
> - Given a reasonable warm start, the critic is expected to give higher Q values for optimal actions for $s \in \mathcal{D} \cap \mathcal{B}_r$ as it learns from data. We aim to leverage this knowledge to enhance the initial teacher policy $\pi_t^\omega$ trained on heuristic domain knowledge. For $s \sim \mathcal{B}$ and $s \models \mathcal{D}$, we calculate the average Q-values over actions suggested by the critic and the teacher, and compare them.
> - If $\mathbb{E}(Q_s^\theta(s,a_s)) > \mathbb{E}(Q_s^\theta(s,a_t))$ , this indicates that the critic expects a higher average return from its action than from the teacher's action. In such cases, we can use the critic’s action to update $\pi_t^\omega$, thereby improving the teacher policy over the domain $\mathcal{D}$. However, solely relying on the critic's Q-values can be misleading, as high Q-values may appear for out-of-distribution (OOD) actions. To prevent the teacher from being updated by OOD actions, we measure the average uncertainty of the Q-values for both the critic and teacher actions.
> - The uncertainty is measured using predictive variance by adding dropout to the Q network over T forward passes. If $\mathbb{E}(Var^T Q_s^\theta(s_r,a_s)) < \mathbb{E}(Var^T Q_s^\theta(s_r,a_t))$ , it suggests that the critic’s actions are learned from expert data in the buffer and are not OOD samples.
>
> Q4 : detach() and target Q
>
> Yes we use .detach() for $Q_{s}^{\theta}(s,a_t)$ treating it as stable target and only $Q_{s}^{\theta}(s,a_s)$ is adjusted. Thank you for pointing this out we have added the definition in revision in line 128.

---

### Author Response · Authors · 2024-11-15
**Summary of rebuttal**

We thank the reviewers for their time and constructive feedback and for highlighting the following strengths.

Strengths:

- The novel concept, intuitive and proper (Reviewer Dypb),
- use of domain knowledge, empirical advantage (Reviewer N2mw, frm3),
- important setting, simple architecture (Reviewer 341n)
- innovative incorporation of domain knowledge, validation on real dataset (Reviewer msTz)
- Easy to read (reviewer frm3)

We acknowledge the areas for improvement suggested by the reviewers and have made concerted efforts to address them. Below, we highlight our responses to the major concerns, followed by detailed responses to individual reviews.

**Simple environment (Reviewer 341n, msTz, N2mw)**

- We want to emphasize that our proposed method was validated on two continuous-domain sales datasets: **Sales Promotion** (business domain) and **Simglucose** (healthcare). These tasks are inherently complex and reflect the intricacies of real-world problems. For a detailed description of these tasks, please refer to [1] and [2].

- While we acknowledge that our methodology relies on domain knowledge—which may be challenging to design for high-dimensional robotics tasks such as those in d4rl (e.g., HalfCheetah)—domain knowledge is often more accessible in business and healthcare contexts. We believe that this type of problem setup deserves research attention, as offline RL is not solely intended for robotics. Its application to real-world business domains is a promising and motivating area of study.

[1] **Simglucose**: Jinyu Xie. Simglucose v0.2.1 (2018) [Online]. Available: [https://github.com/jxx123/simglucose](https://github.com/jxx123/simglucose).

[2] Qin, Rong-Jun, et al. “NeoRL: A near real-world benchmark for offline reinforcement learning.” _Advances in Neural Information Processing Systems_ 35 (2022): 24753-24765.

**Presentation and other issues (341n, Dypb, N2mw, frm3)**

We revised the manuscript as follows:

- Provided a clearer explanation of Eq 2 as follows:
Each decision node $T_{\eta_i}$ has constraint $\phi_{\eta_i}$ that determines its branching, a Boolean indicator $\mu_{\eta_i}$ selects the branch $(\swarrow or \searrow)$ to follow based on whether the constraint $\phi_{\eta_i}$ is satisfied.
- Explained the conditions of Def 4.1 in line 155 as follows:
The two conditions denote 1. some states in $\mathcal{B}$ do not appear in $\mathcal{B}_r$ 2. for certain transitions $\mathcal{B}_r$ contains fewer transition than $\mathcal{B}$.
-  used citep.
- Corrected symbolic inconsistencies in Fig 2.
- Add discussion to how the method is extended to continuous domain as follows:
Furthermore, we extend this to continuous domain by using the regularization in Eq 4 during critic $(Q_s^\theta)$ training for continuous domain and using actions from actor network $(\pi_s)$ for cross entropy loss in Eq 7. (line 273-275).
- Restructured updating teacher section.
- Removed proposition 4.3 due to limited value and moved proposition 4.2 to the appendix stating simplified assumptions.

---

### Note · Authors · 2024-11-25

I have read and agree with the venue's withdrawal policy on behalf of myself and my co-authors.